# FlexSelect: Flexible Token Selection for Efficient Long Video Understanding

**Yunzhu Zhang**[1][*][†] **Yu Lu**[1][†] **Tianyi Wang**[3] **Fengyun Rao**[3]
**Yi Yang**[1,2] **Linchao Zhu**[1,2][‡]
[1]The College of Computer Science and Technology, Zhejiang University
[2]The State Key Lab of Brain-Machine Intelligence, Zhejiang University
[3]WeChat Vision, Tencent Inc.

## Abstract

Long-form video understanding poses a significant challenge for video large language models (VideoLLMs) due to prohibitively high computational and memory demands. In this paper, We propose **FlexSelect**, a flexible and efficient token selection strategy for processing long videos. FlexSelect identifies and retains the most semantically relevant content by leveraging cross-modal attention patterns from a reference transformer layer. It comprises two key components: (1) **a training-free token ranking pipeline** that leverages faithful cross-modal attention weights to estimate each video token's importance, and (2) **a rank-supervised lightweight selector** that is trained to replicate these rankings and filter redundant tokens. This generic approach can be seamlessly integrated into various VideoLLM architectures, such as LLaVA-Video, InternVL and Qwen-VL, serving as a plug-and-play module to extend their temporal context length. Empirically, FlexSelect delivers strong gains across multiple long-video benchmarks – including VideoMME, MLVU, LongVB, and LVBench. Morever, it achieves significant speed-ups (*e.g.,* up to 9 × on a LLaVA-Video-7B model), highlighting FlexSelect's promise for efficient long-form video understanding. Project page: `https://yunzhuzhang0918.github.io/flex_select`.

## 1 Introduction

Long-form video understanding is crucial for applications such as analyzing movies, building multimodal web agents [31], assisting in video surveillance tasks. Recent Video Large Language Models (VideoLLMs) [1, 53, 27, 8, 25, 56, 47, 22, 55] have shown impressive results on short video clips, combining vision and language processing to answer questions or follow instructions about video content. However, extending these models to process long videos presents significant challenges. Long videos yield a substantial volume of visual tokens, often surpassing the context length of transformer-based LLMs. Additionally, processing entire long videos with VideoLLMs leads to excessive computational and memory overhead, making effective long video analysis infeasible.

To mitigate this, some recent works [15, 33] employ training-based compression modules [18, 51] to summarize visual tokens into a compact representation via finetuning on long-video datasets. However, these methods introduce substantial overhead due to additional training. Alternatively, other approaches [13, 35] leverage cross-modal attention scores from pre-trained VideoLLMs to rank

---

[*]Work done during internship at Wechat Vision.

[†]Equal Contribution.

[‡]Corresponding Author.

39th Conference on Neural Information Processing Systems (NeurIPS 2025).

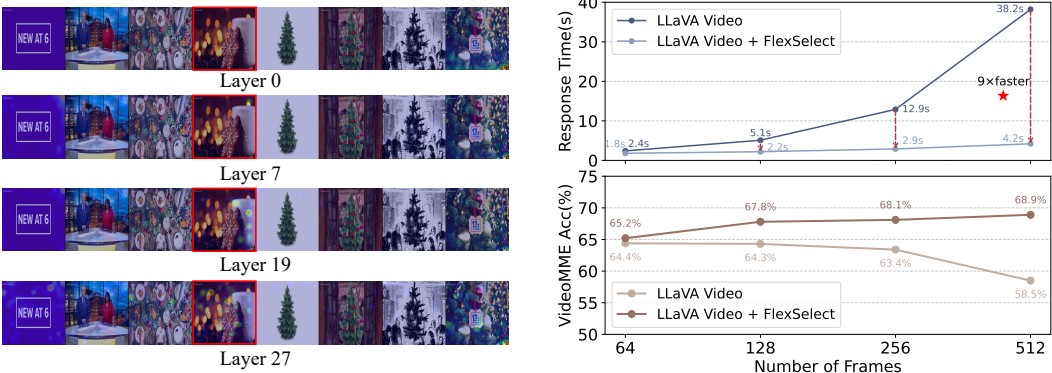

Figure 1: (a) Visualization of cross-modal attention maps of LLaVA-Video-7B across layers (user query : *"what's the color of the cup?"*). Attention scores progressively highlight the query-related regions (the cup) with layer depth, and this highlighting is most pronounced at the specific **reference layer** (layer 19 in example). FlexSelect employs attention scores from this layer to select semantically related visual tokens. (b) VideoMME accuracy and response time (time to generate the first token) of LLaVA-Video-7B. The original model with 64 input frames achieves limited accuracy 64.4% due to inadequate coverage for long video content, while increasing frames will overload the model's context window, reducing accuracy to 58.5% and slowing response time to 38.2s. FlexSelect improves this by filtering irrelevant tokens, achieving 68.9% accuracy at 512 frames with 9× faster response (4.2s).

and prune video tokens without training. While more efficient, these attention-based heuristics often suffer from performance degradation due to inconsistent relevance patterns across transformer layers, which may not reliably reflect the semantic importance of visual tokens for the task.

In this paper, we present **FlexSelect**, a general-purpose framework for efficient long-form video understanding. FlexSelect enables VideoLLMs to **focus on the semanticlly relevant visual tokens for the query** by identifying and filtering out less informative visual tokens before heavy multimodal reasoning occurs. Crucially, FlexSelect is **architecture-agnostic** and does not require any modifications or training of the base VideoLLM. It acts as a preprocessor that significantly extends the model's effective temporal context window without compromising its reasoning ability.

A central observation underlying FlexSelect is that well-trained VideoLLMs inherently encode meaningful cross-modal relevance signals within their internal attention maps. In particular, the cross-attention weights between textual queries and visual tokens progressively reflect semantic alignment across transformer layers and typically peak at an intermediate depth (Fig. 1(a)). Motivated by this, FlexSelect extracts attention scores from an empirically identified *optimal reference layer* to derive **faithful, training-free token importance rankings**. These scores enable the selection of semantically critical visual tokens while discarding redundant or irrelevant ones, thus substantially reducing token volume without compromising the model's reasoning capability. Unlike existing methods that rely on extensive training or oversimplified heuristics, FlexSelect dynamically balances efficiency and performance by exploiting the latent structure of attention patterns within VideoLLMs.

To reduce the computational cost of processing long videos with VideoLLM layers for token ranking, we introduce **a lightweight selector network** trained to mimic the reference layer's ranking. The selector is supervised with a ranking loss and learns to assign higher scores to tokens that the reference model would attend to. This allows for flexible token selection without retraining the full model. Once trained, it efficiently selects the top-ranked visual tokens from long videos, significantly reducing computational overhead. These tokens are then passed into original VideoLLM for final reasoning. In doing so, FlexSelect replicates large model's attention patterns at a fraction of the computational cost.

FlexSelect is a generic approach can be seamlessly integrated into various VideoLLM architectures, such as **LLaVA-Video, InternVL and Qwen-VL**. FlexSelect serves as a plug-and-play inference module requiring no changes to the base VideoLLM. We evaluate it on four challenging long-video understanding benchmarks—VideoMME, MLVU, LongVB, and LVBench—using VideoLLMs of varying sizes, from 7B to 72B parameters. Experimental results show that FlexSelect achieving significant speed-ups (e.g., up to $9\times$ faster inference with LLaVA-Video-7B) while maintaining or

improving performance. Notably, by filtering out irrelevant content, FlexSelect not only accelerates inference but enhances final answer quality.

**Contributions.** To summarize, our contributions are threefold:

- We propose **FlexSelect**, a flexible and architecture-agnostic framework that extracts faithful cross-modal token relevance from an optimal reference layer in VideoLLMs to select semantically important visual tokens for long-form video understanding.
- We design a **lightweight rank-supervised selector** that mimics the cross-modal attention ranking from a reference model layer, enabling fast and accurate token filtering without modifying or retraining the base VideoLLM.
- We demonstrate that FlexSelect is a generic framework applicable to various VideoLLMs, achieving **up to 9× speed-up** and improved accuracy on four long-video benchmarks across multiple model scales.

## 2 Related Works

**Long-form Video Understanding**   Current VideoLLMs showcase remarkable video-language understanding abilities. However, it is still challenge to process long-form videos due to the extensive visual tokens. Recent approaches mainly address this by three ways: (1) applying length extrapolation methods (e.g. YARN [30]) to VideoLLMs and training on longer sequences [7, 49] to support long context input. (2) adopting trainable compression modules to VideoLLM to compress visual content into fewer tokens [11, 21, 33, 15, 14] via post-finetuning. (3) cutting long videos into clips and exploring multi-agent collaboration pipelines to process them [5]. Different from these approaches, we introduce a token selector to directly select semantically relevant visual tokens before LLM generation, without training the large-scale VideoLLM, which is more efficient.

**Attention-based Token Pruning**   Recent works explore visual token pruning for efficient image-text understanding using cross-modal attention scores. FastV [6] first identifies visual token inefficiency in LLM processing, pruning tokens via second-layer attention scores but suffering significant accuracy drops. PyramidDrop[42] observes increasing redundancy with layer depth, applying predefined layer-wise drop ratios to prune more tokens at deeper layers for better results. SparseVLM [52] dynamically adjusts pruning ratios through attention score ranks at each layer, improving performance yet still suboptimal. Recently, FrameFusion [13] and Dycoke [35] analyse redundancy in video data [39], applying similar attention-based token pruning methods to VideoLLMs, improving efficiency but also degrading performance. Meanwhile, several studies [50, 40, 10] show that attention scores fail to reliably indicate semantical relevance of visual tokens because of the attention shift phenomenon, where later tokens tend to have a higher scores due to the autoregressive characteristic of LLM. However, these conclusion are limited because they only discuss layer-averaged attention scores. In this paper, we conduct comprehensive layer-wise analysis on the cross-modal attention pattern, and identify a reference layer where attention scores can reliably indicate the semantical relevance of visual tokens.

## 3 Methods

We present **FlexSelect**, a token selection strategy for long-form video understanding. FlexSelect consists of two complementary components: (1) a **training-free selection pipeline** that leverages faithful cross-modal attention scores in VideoLLM to select semantically relevant visual tokens from a long video, and (2) a **lightweight rank-supervised model** trained to replicate the visual token rankings of the faithful cross-modal attention scores from VideoLLM. This section first analyzes token semantic relevance across transformer layers to identify the reference layer, then explains the training-free FlexSelect procedure for long video understanding, and finally details the rank-supervised token selector, covering its architecture, training objective, and integration into the framework.

### 3.1 Layer-wise Semantic Relevance Analysis

VideoLLMs employ transformer decoder to process video frames as sequences of visual tokens. However, not all tokens are equally relevant to a given query. To identify which decoder layer

best captures **semantic relevance**, we perform a layer-wise analysis using a *"needle-in-haystack"* experiment. Specifically, we insert one unique **needle frame** (image containing distinctive visual content) at random positions in a video sequence. We design a query solely about the needle image so that the visual tokens derived from it are treated as ground-truth *semantically relevant tokens*, while all other tokens are considered irrelevant. By passing the augmented video through the VideoLLM and analyzing the attention patterns at each layer, we assess whether the model successfully highlights these needle visual tokens as semantically aligned with the query.

For a given transformer layer $l$, let $r_i^{(l)}$ denote the *semantic relevance score* of visual token $i$ at that layer. We derive $r_i^{(l)}$ from the model's cross-modal attention. Formally, if $A_{q \to i}^{(l,h)}$ is the attention weight from query tokens $q$ to visual token $i$ in head $h$ of layer $l$, then:

$$r_i^{(l)} = \frac{1}{H} \sum_{h=1}^{H} A_{q \to i}^{(l,h)}, \tag{1}$$

where $H$ is the number of attention heads. This score reflects how strongly the model semantically links visual token $i$ to the query. We rank all visual tokens by $r_i^{(l)}$ in descending order to produce a semantic relevant ranking at that layer.

To quantify how faithfully each layer's semantic relevance scores identify the ground-truth semantically relevant tokens (the needle visual tokens), we use the Recall@K metric, which computes the fraction of relevant tokens recovered in the top-$K$ ranked tokens. Denote TopK$(l)$ is the set of top-$K$ tokens ranked by $r_i^{(l)}$, and $R$ is the set of needle visual tokens considered semantically relevant by construction, then:

$$\text{Recall@}K(l) = \frac{|\text{TopK}(l) \cap R|}{|R|}, \tag{2}$$

We evaluate Recall@$K$ for each transformer layer $l$ by choosing $K = |R|$ (so that perfect recovery of all needle visual tokens in the top-$K$ yields Recall@$K = 1.0$). A higher Recall@$K$ indicates that the layer's relevance is more faithful to the query.

**Results of Layer Analysis:** Figure 2 illustrates the Recall@K value across all transformer layers in the LLM decoder of LLaVA-Video-7B. We observe substantial variation in the effectiveness of different layers at retrieving semantically relevant tokens. Specifically, early layers exhibit relatively low Recall@$K$ scores, suggesting that the attention distributions at these stages are less aligned with the semantic relevance of the query. Very deep layers also not highlight the needle frames as the model has already consolidated the critical visual information into the final token for next token generation. Interestingly, an intermediate layer achieves the highest Recall@$K$, indicating that it best identifies the target visual tokens among its top-ranked outputs. In other words, $L_{\text{ref}}$ serves as the optimal attention layer that most reliably captures truly semantic-relevant tokens in the sequence. Based on this finding, we designate layer $L_{\text{ref}}$ as the reference layer for guiding token selection in FlexSelect. All subsequent token semantic-relevance computations in our method will use the reference layer's attention scores as the measure-

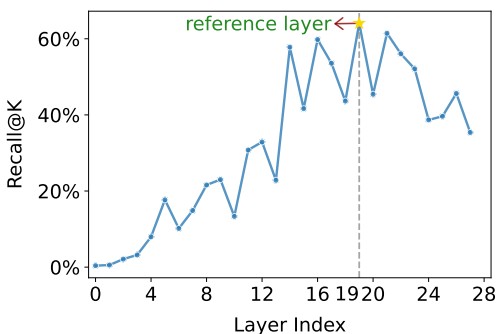

Figure 2: Recall@K values across different layers in LLaVA-Video-7B. Recall@K metric is the recall ratio of ground-truth relevant tokens (e.g., needle-frame tokens) among the top-K tokens ranked by a layer's cross-modal attention scores. A higher Recall@K indicates the attention scores of that layer can more accurately identify the semantically related visual tokens. We choose the optimal layer with the highest Recall@K as the reference layer for token selection.

ment of semantic importance. More analysis including more base models and PCA visualization on tokens can be found in appendix A.1.

### 3.2 Training-Free FlexSelect Pipeline

Even with the reference layer $L_{\text{ref}}$ identified, processing a long video in a single forward pass is typically infeasible due to the quadratic cost of self-attention and memory constraints. To address

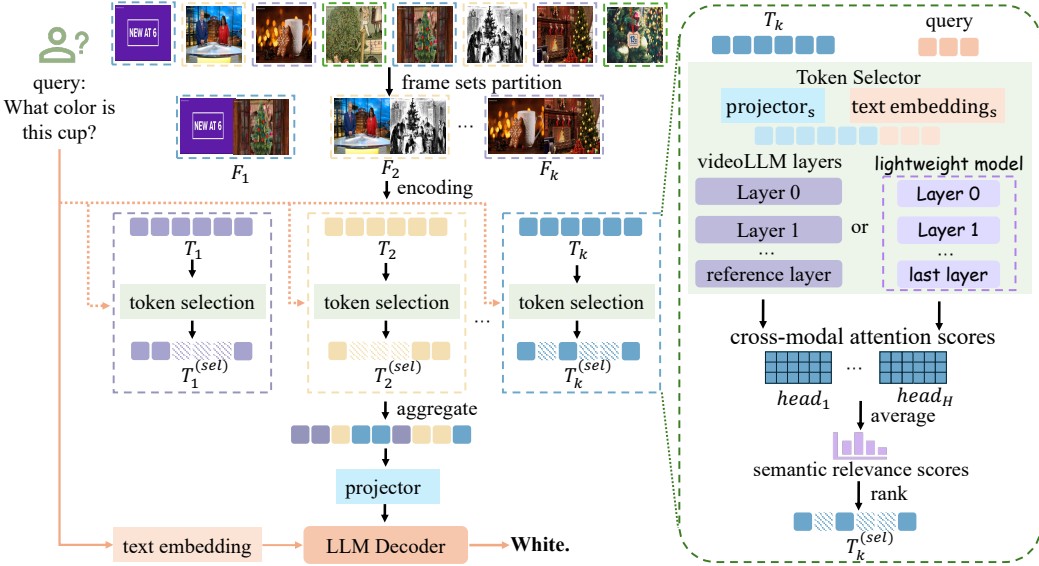

Figure 3: **Overview of FlexSelect token selection pipeline.** Given a long video and a query, FlexSelect first partitions the video into frame sets and encodes each into visual tokens. For each set, a token selector identifies semantically relevant tokens by ranking cross-modal attention scores from a reference layer in a pre-trained VideoLLM or a lightweight selector network trained to approximate it. In this process, the projector$_s$ and text embedding$_s$ are employed to convert the visual tokens and user queries into tokens that match the dimension of subsequent transformer layers. After getting the scores, the top-ranked tokens across all segments are aggregated and projected into the decoder for final reasoning. FlexSelect operates in a training-free or rank-supervised mode, and serves as a plug-and-play module that enables efficient long-video understanding without requiring modifications to the base VideoLLM.

this, we propose a training-free FlexSelect strategy that enables efficient visual token selection while preserving semantic coverage across the entire video. The video is divided into multiple *frame sets* uniformly, with token selection performed independently within each set. This approach ensures comprehensive temporal coverage without requiring the entire video sequence to be processed at once.

**Frame Sets Partition.** We partition the extensive frames into frame sets and each set most contains $S$ frames to prevent token number exceeding the VideoLLM's maximum context length. Given a video with $N$ sample frames $\{f_1, f_2, \ldots, f_N\}$, we construct $K = \lceil N/S \rceil$ *frame sets* $\{\mathcal{F}_1, \mathcal{F}_2, \ldots, \mathcal{F}_K\}$ such that each set samples frames at a same stride. Formally, the $j$-th frame set is defined as $\mathcal{F}_j = \{f_i \mid i \equiv j \mod N\}$, where $j \in \{1, \ldots, K\}$. This sampling ensures that each frame set spans the entire video with different temporal offsets, capturing diverse temporal dynamics and reducing redundancy between sets.

Each frame set $\mathcal{F}_j$ is encoded by the VideoLLM's visual encoder. This yields a sequence of visual tokens $T_j = \{t_{j,1}, \ldots, t_{j,M}\}$ for each set, $M$ means the total number of visual tokens in a frame set.

**Semantic Relevance Scoring and Token Selection.** Within each frame set $\mathcal{F}_j$, we compute a semantic relevance score $r_{j,i}$ for each token $t_{j,i}$ using the attention mechanism at layer $L_{\text{ref}}$:

$$r_{j,i} = \frac{1}{H} \sum_{h=1}^{H} A_{q \to t_{j,i}}^{(L_{\text{ref}}, h)},$$

where $A_{q \to t_{j,i}}^{(L_{\text{ref}}, h)}$ is the attention weight from the query tokens $q$ to token $t_{j,i}$ in head $h$, and $H$ is the number of heads. This relevance score reflects the semantic alignment between each token and the query. We then rank the tokens in each $T_j$ by $r_{j,i}$ and select the top-$k$ tokens $T_j^{(\text{sel})} = \text{TopK}_k (\{r_{j,i}\})$, yielding a set of semantically relevant tokens for each frame set.

**Aggregation and Final Token Composition.** The selected tokens from all frame sets are aggregated to form the final visual token input $T_{\text{selected}} = \bigcup_{j=1}^{K} T_j^{(\text{sel})}$. This merged token set provides a globally informed yet compact representation of the video.

By constructing $K$ frame sets with uniform sampling and processing them independently, our proposed FlexSelect strategy ensures that the framework scales efficiently to long video sequences. The method minimizes computational overhead by leveraging the parallelizability of processing smaller frame sets, while maintaining temporal fidelity across the video.

### 3.3 Rank-Supervised Lightweight Token Selector

While the training-free approach described above effectively reduces computational overhead, it still relies on partial forward passes through the large VideoLLM to score visual tokens. To further enhance inference efficiency, we introduce a *lightweight token selector* trained via rank supervision to predict semantic relevance scores independently. The selector model is explicitly designed to replicate the token-ranking behavior observed at the reference transformer layer $L_{\text{ref}}$.

**Architecture and Input.** Our lightweight token selector is a compact, shallow transformer-based network intended to substantially reduce inference costs compared to the large-scale VideoLLM. The model receives two inputs: visual tokens extracted by the vision encoder and the corresponding textual query. It outputs predicted semantic relevance scores for each visual token in the input sequence. Formally, for a visual token sequence $\{t_1, t_2, \ldots, t_M\}$ paired with a textual query $q$, the selector produces scores $\{\hat{r}_1, \hat{r}_2, \ldots, \hat{r}_M\}$ indicating each token's predicted semantic alignment with the query. These scores are subsequently used to select the most relevant visual tokens, similar to the training-free method described above.

To leverage pretrained knowledge and accelerate training convergence, we initialize the selector from a smaller-scale pretrained VideoLLM (approximately 0.5B parameters). In our experiments, we separately train selectors for LLaVA-Video-7B, Qwen2.5VL-7B, and InternVL2.5-8B.

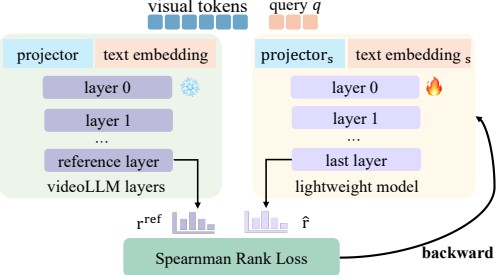

Figure 4: Illustration of our rank-supervised training. We align lightweight model's predicted scores $\hat{\mathbf{r}}$ with the reference layer's sematic relevance scores $\mathbf{r}^{\text{ref}}$ by optimizing the spearman rank correlation coefficient between them. Once trained, the ranking derived from these two scores will follow similar order, enabling the lightweight model to rank the visual tokens as the reference layer does and select the related tokens quickly.

**Rank-Supervised Training Objective.** We train the lightweight selector with rank supervision, directly leveraging semantic relevance rankings provided by the reference transformer layer $L_{\text{ref}}$ from the larger VideoLLM. For each training video-query pair, we first compute the semantic relevance scores $\mathbf{r}^{\text{ref}} = [r_1^{\text{ref}}, r_2^{\text{ref}}, \ldots, r_M^{\text{ref}}]$ at the reference layer $L_{\text{ref}}$. Then, the lightweight selector predicts scores $\hat{\mathbf{r}} = [\hat{r}_1, \hat{r}_2, \ldots, \hat{r}_M]$ for the same tokens.

The training goal is to align the predicted semantic relevance ranking $\hat{\mathbf{r}}$ with the reference ranking $\mathbf{r}^{\text{ref}}$. To quantify this alignment, we use the spearman rank correlation coefficient [34]:

$$\rho_{\text{spearman}}(\mathbf{r}^{\text{ref}}, \hat{\mathbf{r}}) = \frac{\sum_{i=1}^{M}(\text{rank}(r_i^{\text{ref}}) - \overline{\text{rank}(r^{\text{ref}})})(\text{rank}(\hat{r}_i) - \overline{\text{rank}(\hat{r})})}{\sqrt{\sum_{i=1}^{M}(\text{rank}(r_i^{\text{ref}}) - \overline{\text{rank}(r^{\text{ref}})})^2 \sum_{i=1}^{M}(\text{rank}(\hat{r}_i) - \overline{\text{rank}(\hat{r})})^2}}. \tag{3}$$

$\rho_{\text{spearman}}(\mathbf{r}^{\text{ref}}, \hat{\mathbf{r}})$ closer to 1 indicates stronger consistency between the ranking orders of $\mathbf{r}^{\text{ref}}$ and $\hat{r}$, while closer to 0 suggests no monotonic relationship. The corresponding training loss is defined as:

$$\mathcal{L}_{\text{rank}} = 1 - \rho_{\text{spearman}}(\mathbf{r}^{\text{ref}}, \hat{\mathbf{r}}). \tag{4}$$

| Model | Size | VideoMME | | MLVU | LongVB | LVBench |
|---|---|---|---|---|---|---|
| | | Long | Overall | M-Avg | Val | Test |
| **Proprietary Models** | | | | | | |
| GPT-4o [29] | - | 65.3 | 71.9 | 64.6 | 66.7 | 34.7 |
| Gemini-1.5-Pro [36] | - | **67.4** | **75.0** | - | 64.0 | 33.1 |
| **Open-Source VideoLLMs** | | | | | | |
| LongVU [45] | 7B | 50.1 | 59.3 | 63.7 | 52.1 | 43.5 |
| mPLUG-Owl3 [45] | 7B | 50.1 | 59.3 | 63.7 | 52.1 | 43.5 |
| NVILA [25] | 8B | 54.8 | 64.2 | 70.1 | 57.7 | - |
| VideoLLaMA3 [46] | 7B | - | 66.2 | 73.0 | 59.8 | 45.3 |
| Aria [17] | 8x3.5B | 58.8 | 67.6 | 70.6 | 65.3 | - |
| Oryx-1.5 [26] | 34B | 59.3 | 67.3 | 72.3 | 62.0 | 30.8 |
| Video-XL-Pro [24] | 3B | - | 60.0 | 70.6 | 56.7 | - |
| LongVU [32] | 7B | 59.5 | 60.6 | 65.4 | - | - |
| SF-LLaVA-1.5 [43] | 7B | - | 63.9 | 71.5 | 62.5 | 45.3 |
| TPO [19] | 7B | 55.4 | 65.6 | 71.1 | 60.1 | - |
| Quato [28] | 7B | 55.7 | 65.9 | 71.9 | 59.0 | - |
| ViLAMP [9] | 7B | 57.8 | 67.5 | 72.6 | 61.2 | 45.2 |
| VideoChatFlash [20] | 7B | 55.4 | 65.3 | 74.7 | 64.7 | 48.2 |
| LLaVA-Video [53] | 7B | 52.9 | 64.4 | 68.6 | 58.2 | 43.1 |
| + FlexSelect | 7B | 59.8 ↑6.9 | 68.9 ↑4.5 | 73.2 ↑4.6 | 61.9 ↑3.7 | 52.9 ↑9.8 |
| + FlexSelect-Lite | 7B | 58.3 ↑5.4 | 68.3 ↑3.9 | 71.8 ↑3.2 | 60.7 ↑2.5 | 52.2 ↑9.1 |
| InternVL2.5 [8] | 8B | 52.8 | 64.2 | 68.9 | 59.5 | 43.4 |
| + FlexSelect | 8B | 58.1 ↑5.3 | 67.0 ↑2.8 | 71.9 ↑3.0 | 60.1 ↑0.6 | 49.7 ↑6.3 |
| + FlexSelect-Lite | 8B | 57.9 ↑5.1 | 67.2 ↑3.0 | 71.9 ↑3.0 | 61.2 ↑1.7 | 49.9 ↑6.5 |
| Qwen2.5-VL [1] | 7B | 55.6 | 65.4 | 70.2 | 59.5 | 45.3 |
| + FlexSelect | 7B | 59.3 ↑3.7 | 68.2 ↑2.8 | 72.5 ↑2.3 | 62.4 ↑2.9 | 51.2 ↑5.9 |
| + FlexSelect-Lite | 7B | 58.6 ↑3.0 | 67.4 ↑2.0 | 70.3 ↑0.1 | 61.9 ↑2.4 | 50.0 ↑4.7 |
| LLaVA-Video [53] | 72B | 61.9 | 70.0 | 71.2 | 62.4 | 45.5 |
| + FlexSelect | 72B | 66.1 ↑4.2 | 73.1 ↑3.1 | 76.0 ↑4.8 | **66.9** ↑4.5 | 55.5 ↑10.0 |
| Qwen2.5 VL [1] | 72B | 63.9 | 73.4 | 76.3 | 66.2 | 47.3 |
| + FlexSelect | 72B | 66.9 ↑3.0 | 74.4 ↑1.0 | **76.6** ↑0.3 | 66.4 ↑0.2 | **56.6** ↑9.3 |

Table 1: Comprehensive evaluation on different long video benchmarks. Gray rows show baseline results reproduced from public model weights. FlexSelect employs attention scores from the reference layer in VideoLLM for token selection, while FlexSelect-Lite utilizes scores from our lightweight token selector. Our methods consistently improve performance when integrated into various VideoLLMs by selecting semantically relevant visual tokens from extensive sampled frames. The implementation details of these results can be found in appendix A.4.

Since the ranking operation (i.e. argsort) itself is non-differentiable, we adopt a differentiable sorting algorithm [2] to approximate ranks and enable gradient backpropagation. This ensures effective training while strictly supervising the model on ranking quality.

**Integration into FlexSelect Pipeline** At inference, the lightweight selector directly replaces the transformer layers from the bigger VideoLLM to process visual tokens and query inputs and produces semantic relevance scores, eliminating the need for intermediate VideoLLM computation. By using these scores to select only the top-ranking tokens, we significantly reduce computational overhead, enabling efficient long-video understanding at scale. For clarity, we denote the FlexSelect integrated with the lightweight token selector as FlexSelect-Lite.

## 4 Experiments

### 4.1 Models and Benchmarks

We conduct comprehensive evaluations across diverse VideoLLM architectures and scales to assess the generalizability of our FlexSelect, including: (1) LLaVA-Video (7B/72B) [53], (2) InternVL-2.5

8B [8], and (3) Qwen2.5VL (7B/72B) [1]. The models are evaluated on four established long-video understanding benchmarks: (1) LongVideoBench [41], a benchmark designed for accurate retrieval and reasoning in long-context videos, we report the validation set result. (2) MLVU [54], a multitask benchmark specifically designed for long-form video understanding, we report the M-avg score. (3) VideoMME [12], a comprehensive evaluation across short/medium/long videos, we report the the long and overall results without subtitles. (4) LVBench [38], an extreme-long video benchmark with average video length reaches to one hour.

## 4.2 Main Results

**Compared to SoTAs**   As shown in Table 1, our methods consistently enhance VideoLLM performance across multiple benchmarks. For LLaVA-Video-7B, FlexSelect delivers a +5.5 points improvements in average (+4.5 on VideoMME, +4.6 on MLVU, +3.7 on LongVB, and +9.8 on LVBench), surpassing other long-form video understanding methods like SF-LLaVA [43], TPO [19], Quato [28] and ViLAMP [9] at 7B parameter scale, demonstrating its effectiveness for long video understanding. FlexSelect-Lite maintains most of these gains (+3.9 in average) with less computational cost, validating the effectiveness of our rank-supervised token selector. The method's adaptability is further confirmed by similar improvements when integrated into other models: Qwen2.5-VL-7B shows an average gain of +3.5 with FlexSelect and +2.3 with FlexSelect-Lite, while InternVL2.5-8B achieves an average improvement of +3.2 with FlexSelect and +3.7 with FlexSelect-Lite.

For larger-scale models, FlexSelect continues to deliver impressive results: LLaVA-Video-72B with FlexSelect shows a +4.5 improvement on LongVB (62.4 $\rightarrow$ 66.9), outperforming GPT-4o (66.7). Qwen2.5VL-72B with FlexSelect sets new state-of-the-art results among open-source methods, achieving +9.3 on LVBench (47.3 $\rightarrow$ 56.6). These consistent improvements across model sizes and benchmarks demonstrate the ability of FlexSelect to select semantically relevant tokens, confirming the effectiveness of our token selection mechanism.

**Compared to other Token Pruning Methods**   We compare FlexSelect with FastV [6], FrameFusion [13], Dycoke [13] and VisionZip [44] using the same base model. As shown in Table 2, when sampling 32 or 64 frames, FlexSelect achieves better performance than these methods with the same retain ratio. When sampling 512 or 1024 frames, FrameFusion, Dycoke and VisionZip all encounter Out of Distribution (OOD) issues. FlexSelect, on the other hand, can effectively process these long input frames due to its Frame Sets Partition and Token Selection operation and achieves significantly better results.

| Method | Sample Frames | Retain Ratio (%) | VideoMME (%) |
|---|---|---|---|
| LLaVA-Video-7B | 64 | 100.00 | 64.4 |
| + FrameFusion | 64 | 30.00 | 61.3 |
| + FlexSelect | 64 | 30.00 | **64.9** |
| + FrameFusion | 512 | 6.25 | OOM |
| + FlexSelect | 512 | 6.25 | **68.9** |
| LLaVA-OV-7B | 32 | 100.00 | 58.5 |
| + FastV | 32 | 35.00 | 57.3 |
| + DyCoke | 32 | 14.25 | 58.3 |
| + FlexSelect | 32 | 14.25 | **60.4** |
| + DyCoke | 512 | 6.25 | OOM |
| + FlexSelect | 512 | 6.25 | **63.7** |
| Qwen2.5-VL-7B | 64 | 100.00 | 63.6 |
| + VisionZip | 64 | 50.00 | 62.4 |
| + FlexSelect | 64 | 50.00 | 62.6 |
| + VisionZip | 1024 | 50.00 | OOM |
| + FlexSelect | 1024 | 6.25 | 68.2 |

Table 2: Comparison of FlexSelect with other token reduction methods on the VideoMME.

**Efficient Long Video Understanding**   Our method significantly improves VideoLLM's efficiency when processing long videos with extensive frames. We evaluate the response time (i.e., time cost to generate the first token) on LLaVA-Video-7B. As shown in Figure 5, both FlexSelect and

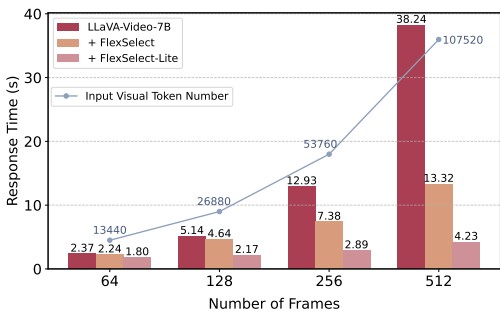

Figure 5: Response time when sampling different number of frames. FlexSelect accelerates inference by selecting semantically relevant visual tokens faithfully.

Figure 6: VideoQA Acc with different layers for token selection. The reference layer (*19th*) yields a more faithful cross-modal ranking for token selection.

| Input Frames | VideoMME (%) |
|---|---|
| 64 | 65.2 |
| 128 | 67.8 |
| 256 | 68.1 |
| 512 | **68.9** |
| 1024 | 68.1 |

| Max Selected Tokens | VideoMME (%) |
|---|---|
| 1,680 | 67.1 |
| 3,360 | 68.4 |
| 6,720 | **68.9** |
| 13,440 | 68.1 |

Table 3: Ablation on input frames and max selected tokens of training-free FlexSelect. Evaluations are conducted on LLaVA-Video-7B.

| Training Data Scale | VideoMME (%) |
|---|---|
| No training | 61.9 |
| 1% (14k samples) | 63.3 |
| 2% (29k samples) | 63.4 |
| 5% (67k samples) | **63.9** |
| 10% (134k samples) | 63.5 |

| Instruction Type | VideoMME (%) |
|---|---|
| Detail caption | 63.5 |
| Open-ended QA | 63.6 |
| Multi-choice QA | 63.7 |
| Mixed types | **63.9** |

Table 4: Ablation on data scale and instruction type of rank-supervised training. Our token selector boosts performance while requiring only 5% of training data.

FlexSelect-Lite reduce response time for the same number of frames by decreasing the visual token number. FlexSelect-Lite achieves more pronounced acceleration, highlighting the efficiency of our lightweight token selector. This speed advantage grows progressively with more sampled frames: for 512 frames, LLaVA-Video-7B requires 38.24 s per sample, while FlexSelect-Lite reduces this to just 4.23 s—achieving a 9× speedup. More analysis on FLOPs estimation can be found in appendix A.3.

## 4.3 Ablations

**Effectiveness of Reference Layer**  We compare the performance of FlexSelect on LLaVA-Video-7B when using attention scores from different layers for token selection. The evaluation is conducted with 512 input frames and 6,720 max selected tokens. As shown in Figure 6, the reference layer (layer 19) performs the best, as it can more accurately select semantically related tokens.

**Influence of Input Frames and Max Selected Tokens**  We evaluate LLaVA-Video-7B with FlexSelect on VideoMME under varying input frames and max selected tokens. As demonstrated in Table 3, when fixing the max selected tokens to 6,720, we observe progressive accuracy improvements as the input frames increase from 64 to 512, followed by a slight degradation at 1,024 frames. Similarly, with input frames fixed at 512, performance improves when increasing max selected tokens from 1,680 to 6,720, but further selecting 13,440 tokens reduces the accuracy. The result shows that insufficient input frames or selected tokens cannot adequately cover the video content, risking overlook critic infromation, while excessive frames or tokens may introduce semantically irrelevant noise - both scenarios ultimately degrading performance.

**Scales and Types of Training Data**  We train token selectors for LLaVA-Video-7B under different data scale and video instruction types, and evaluate them with 64 input frames and 1,680 max selected

tokens. We first randomly sample 4 subsets (1%, 2%, 5%, 10%) from LLaVA-Video-178K [53] without considering video instruction type, using them as training datasets to train different token selectors. As shown in Table 4, directly initializing the token selector from a small-scale VideoLLM achieves an accuracy of 61.9%, while after training with just 1% of the data, the accuracy rises to 63.3%. The performance peaks when trained on 5% of the data (approximately 67k samples), and saturates with more data, demonstrating the quick convergency and training efficiency of our rank-supervised training. Subsequently, we fix the data scale at 67k samples while varying the video instruction types. Results show comparable performance across multiple-choice QA, open-ended QA, and video captioning data, suggesting that our training is effective on various instruction types.

| Token Selector Params | Response Time | Datasets | | | |
|:---:|:---:|:---:|:---:|:---:|:---:|
| | | VideoMME | LongVB | MLVU | LVBench |
| 0.5 B | 4.0 s | 67.2 | 61.2 | 71.9 | 49.9 |
| 1.8 B | 7.4 s | 67.3 | 60.6 | 71.7 | 49.7 |
| 3.0 B | 12.0 s | 67.4 | 61.8 | 72.8 | 50.1 |

Table 5: Ablation on token selector parameters of rank-supervised training. We train different parameter size token selector for InternVL2.5-8B and test their acuracy on four benchmark. Evaluations are conducted with input frames set to 512 and max selected tokens set to 8,256.

**Parameter Scale of Token Selector**  We investigate the impact of token selector parameter scale on performance. We train token selectors with 0.5B, 1.8B, and 3B parameters for InternVL2.5-8B, which are initialized from InternVL2.5-1B, InternVL2.5-2B and InternVL2.5-4B respectively, then we compare their performance on four benchmarks. Our results in Table 5 show that the 3B token selector achieves slightly higher scores than others but incurs significant computational overhead, and the 0.5B model delivers comparable performance while requiring only one-third of the response time of 3B model(4.0s vs. 12.0s). This indicates that scaling parameter of token selector brings limited gains, and 0.5B is a cost-effective choice.

**Compared to Majority Voting**  We compare FlexSelect against the majority voting method to verify that the performance gains stem not merely from processing more frames but from the mechanism that locating the query-related visual tokens. For majority voting, we divide each video into 64 temporal bins, randomly sample one frame per bin, and repeated this process 8 times. The final answer is determined by the most frequent prediction. As shown in Table 6, majority voting not only requires more time (18.96 seconds) but also achieves significantly lower performance (64.9) compared to FlexSelect (68.9). This result strongly demonstrates that simply increasing the number of processed frames and smoothing noise via ensemble voting is far less effective than FlexSelect, which enables global consideration and fusion of key information across segments.

| Method | Total Sampled Frames | Time Cost | VideoMME |
|:---:|:---:|:---:|:---:|
| LLaVA-Video-7B | 64 | 2.37 s | 64.4 |
| Majority Voting | 64 × 8 | 18.96 s | 64.9 |
| FlexSelect | 512 | 13.32 s | **68.9** |

Table 6: Comparison between FlexSelect and the Majority Voting.

## 5  Conclusion

This paper presents FlexSelect, a flexible and efficient token selection method that leverages cross-modal attention scores in VideoLLMs to identify query-relevant visual tokens. Our approach combines: (1) training-free attention-based token ranking, and (2) a lightweight selector for fast filtering. Its architecture-agnostic design enables integration into diverse VideoLLMs without modification. By selecting the most query-relavant visual tokens, FlexSelect achieves SoTA results on VideoMME (74.4), MLVU (76.6), LongVideoBench (66.9), and LVBench (56.6) while reducing tokens by over 90% (up to 9× speedup). This method demonstrates powerful long-form video understanding capabilities, enabling effective analysis of long-form videos with minimal computational overhead.

# 6 Acknowledgements

This work is partially supported by Major program of the National Natural Science Foundation of China (T2293720/T2293723). This work was supported in part by "Pioneer" and "Leading Goose" R&D Program of Zhejiang (No. 2025C02032), the Fundamental Research Funds for the Central Universities (226-2025-00055). This work was also supported by the Earth System Big Data Platform of the School of Earth Sciences, Zhejiang University.

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

# A Appendix / supplemental material

## A.1 Details of Recall@K Experiment

**Needle Images and Queries** The needle images and corresponding queries used in our Recall@K experiment are directly borrowed from the V-NIAH experiment in LongVA [49], which provides five needle-query pairs. We randomly sample 128 videos from the VideoMME [12] test set and insert each needle-query pair into them, resulting in a total of 640 test samples. We compute Recall@K on these samples.

**Recall@K of Various Models** In addition to LLaVA-Video-7B, we calculate Recall@K for LLaVA-Video-72B, InternVL2.5-8B, Qwen2.5VL-7B and Qwen2.5VL-72B. The result is shown at Figure 7. We identify the reference layer of these models: layer 15 for InternVL2.5-8B, layer 60 for LLaVA-Video-7B, layer 20 for Qwen2.5VL-7B and layer 60 for Qwen2.5VL-72B.

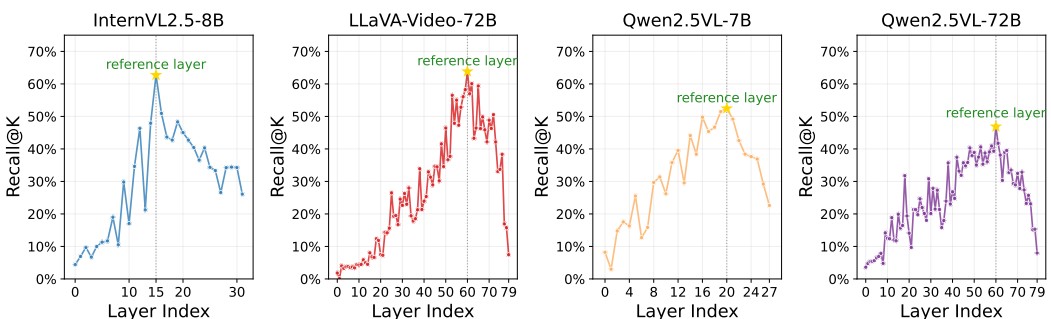

Figure 7: Recall@K across layers of different VideoLLMs.

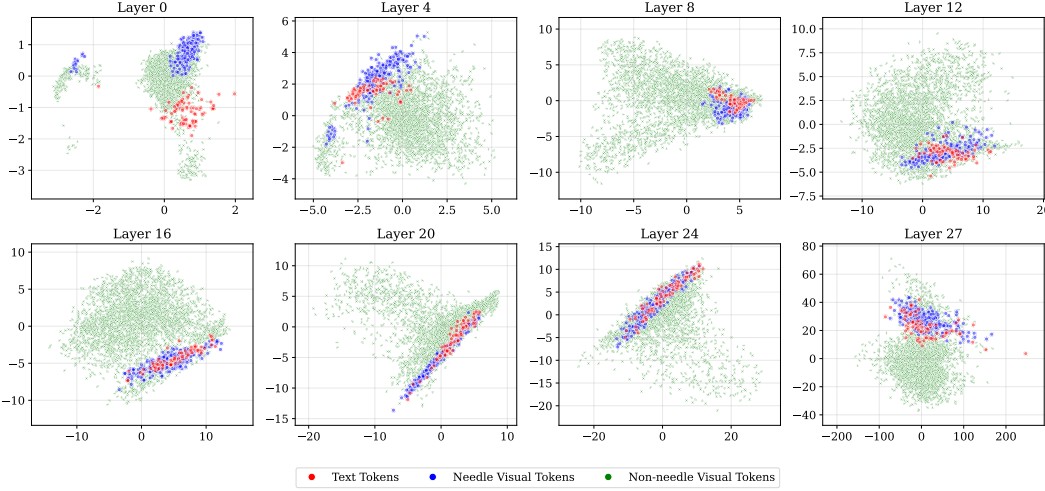

Figure 8: PCA visualization of query tokens, needle visual tokens(i.e. semantically related tokens) and non-needle visual tokens from different layers of LLaVA-Video-7B. We found that the correlation between destributions of text tokens and needle visual tokens varies with layer depth, matching the trend of Recall@K: at shallow layers, their distributions diverge; in intermediate layers, a strong linear correlation emerges; while at very deep layers, this linear correlation weakens. This finding demonstrates that semantic alignment between visual and text tokens established at intermediate layers in VideoLLMs, which further explains the emergence of the reference layer.

**PCA Visualization** We further visualize the query tokens, needle visual tokens and non-needle visual tokens using Principal Component Analysis (PCA) [3] to examine cross-modal semantic alignment across different layers in VideoLLMs. As shown in Figure 8 (LLaVA-Video-7B), a

significant distribution discrepancy exists between text tokens and needle visual tokens in shallow layers. As the layer depth increases, these distributions develop a strong linear correlation in deeper layers, which subsequently weakens in the deepest layers. This indicates that the cross-modal semantic alignment is initially absent in early layers, gradually strengthens in intermediate layers, and then decreases in the final layers, which is consistent with the trend of Recall@K.

## A.2 More Evaluation Results

**Compared to Frame Selection Method**  We evaluate and compare FlexSelect with BOLT [23] on the Video-MME benchmark. BOLT enhances long-form video understanding by selecting key frames, while FlexSelect operates at the token level. We sample at 1 fps (max to 512 frames) and select 32 * 196 tokens per video, following the same setting as BOLT. As shown in Table 7, the result demonstrates that FlexSelect with fine-grained token-level selection significantly surpasses BOLT that with frame-level selection.

| Method | Sample Frames | Visual Token Number | Video-MME |
|---|---|---|---|
| LLaVA-OV-7B | 32 | 32 * 196 | 58.5 |
| + BOLT | 512 | 32 * 196 | 59.9 |
| + FlexSelect | 512 | 32 * 196 | 63.7 |

Table 7: Comparision with Frame Selection Methods.

**Performance on Caption Tasks**  VideoDC [4] is a detailed video captioning benchmark that evaluates the capability of models for caption generation. We compare results of VideoDC with and without FlexSelect on LLaVA-OV-7B. As shown in Table 8, the result indicates that FlexSelect achieves comparable performance with the original model with 23.25% visual tokens retained. When models are prompted with queries like "Describe the video in detail", the reference layer's attention patterns still exhibit meaningful selectivity, focusing on important elements like main subjects and actions while deemphasizing unnecessary background. The cross-modal attention mechanism inherently adjusts its selection strategy based on query type—focusing narrowly for specific questions while maintaining broader coverage for descriptive tasks. This indicates that FlexSelect's semantic relevance scoring effectively handles various video understanding scenarios. More Visualization can be seen at Figure 9.

| Methods | Sample Frames | Retain Ratio (%) | VideoDC Score |
|---|---|---|---|
| LLaVA-OV-7B | 32 | 100.00 | 3.30 |
| + FlexSelect | 32 | 23.25 | 3.29 |

Table 8: Performance on Caption Tasks.

**Performance on Open-ended Tasks**  VideoEvalPro [24] is a benchmark for open-ended video query answering, with an average video length of 38.25 minute. We compare the results of VideoEval-Pro with and without FlexSelect on different models. As shown in Table 9, these results demonstrate that FlexSelect consistently improves performance on open-ended tasks, validating its generalization beyond multi choice tasks.

| Methods | Sample Frames | Retain Ratio (%) | VideoEvalPro |
|---|---|---|---|
| LLaVA-Video-7B | 64 | 100.00 | 23.4 |
| + FlexSelect | 512 | 6.25 | 30.7 |
| Qwen2.5VL-7B | 512 | 100.00 | 21.3 |
| + FlexSelect | 1024 | 6.25 | 25.2 |
| InternVL2.5-8B | 64 | 100.00 | 21.7 |
| + FlexSelect | 512 | 6.25 | 28.2 |

Table 9: Performance on Open-ended Tasks.

### A.3 FLOPs Analysis

Suppose the transformer decoder of VideoLLM has $L$ layers, $h$ heads, and the hidden states size is $d$, the intermediate size of FFN is $m$. For an input sequence of $n$ tokens, the FLOPs of the prefilling stage can be estimated as:

$$\text{FLOPs} = L \times (4nd^2 + 2n^2 d + 2ndm) \approx 2Ln^2 d,$$

since $n \gg d$ and $n \gg m$ in typical scenarios.

When using FlexSelect, assuming the $M$-th layer serves as the reference layer and the input sequence is partitioned into $K$ segments via our frame set partition operation, ultimately selecting $n'$ tokens, the FLOPs can be estimated as:

$$\text{FLOPs'} = M \times \left(4nd^2 + \frac{2n^2 d}{K} + 2ndm\right) + L \times (4n'd^2 + 2n'^2 d + 2n'dm) \approx \frac{2Mn^2 d}{K},$$

since in our configuration, $n' = 0.0625n$, making the $L \times (4n'd^2 + 2n'^2 d + 2n'dm)$ term negligible in FLOPs'. Consequently, FlexSelect requires only $\frac{M}{L} \times \frac{1}{K}$ of the original FLOPs.

Similarly, when employing FlexSelect-Lite, the Flops can be estimated as $\frac{2L'n'^2 d'}{K}$, where $L'$ and $d'$ are layer num and hidden states dimension of lightweight token selector. This FLOPs is more smaller than FlexSelect because $L' < L$ ans $d' < d$. FLOPs estimation only provides a theoretical reference for computational cost. For practical considerations, we recommend referring to the actual time cost analysis in Figure 5.

### A.4 Implementation Details

The evaluations in main Table 1 are conducted under LMMS-Eval [48] framework on 8 96G H20 GPUs. We set the input frames number $N$ to 1024, 512, 512 for Qwen2.5VL(7B/72B), LLaVA-Video(7B/72B), and InternVL2.5-8B respectively, and max subset frames number $S$ to 64 for all models. Denoting $N_{\text{image}}$ is the token number required for encoding one frame, we select 7,010 ($64 * N_{\text{image}}$), 6,720 ($32 * N_{\text{image}}$), 8,256 ($32 * N_{\text{image}}$) visual tokens for these models respectively, which is 6.25% of original input length. The reference layer $L$ for token selection are set to Layer 15 for InternVL2.5-8B, Layer 19 for LLaVA-Video-7B, Layer 20 for Qwen2.5VL-7B, Layer 60 for LLaVA-Video-72B and Qwen2.5VL-72B determined by $\arg\max_L \text{Recall@}K(L)$.

We train lightweight token selectors for 7B/8B models but exclude 72B due to memory constraints. For token selector of LLaVA-Video-7B, we initialize it with the decoder of LLaVA-OneVision-0.5B [16]; for token selector of InternVL2.5-8B, we initialize it with the decoder of InternVL-1B. In the case of Qwen2.5VL-7B, where no similarly-sized VideoLLM is available, we directly initialize it with Qwen2.5-Instruct-0.5B [37]. We randomly select a small ( 5%) subset of LLaVA-Video-178K [53] as the training data, which contains about 67k video instruction samples. We uniformly sample 64 frames from each video during selector training. Token selectors for LLaVA-Video and InternVL2.5 were trained for 1 epoch, while the token selector for Qwen2.5VL was trained for 3 epochs since it was initialized from a language model, which is lack of visual prior knowledge, and requires more training steps to achieve convergence.

### A.5 Limitations

FlexSelect enhances VideoLLMs' long-video understanding by selecting semantically relevant tokens, without modifying or retraining the base VideoLLM. Thus, its performance ceiling is bounded by the host VideoLLM's native capabilities.

FlexSelect-Lite trains a lightweight token selector to predict the reference layer's importance scores in large-scale VideoLLMs. While more efficient, it typically underperforms direct reference-layer token selection. Nevertheless, compared to other heavily trained alternatives and existing sub-optimal token pruning methods, FlexSelect remains an efficient and effective solution for boosting diverse VideoLLMs' long-video comprehension.

## A.6 Social Impacts

FlexSelect operates as a preprocessing module that selects semantically relevant visual tokens for diverse VideoLLMs. It is important to note that this method cannot prevent the base VideoLLMs from potentially generating erroneous, biased, or harmful hallucinations.

## A.7 Visualization of Some Examples

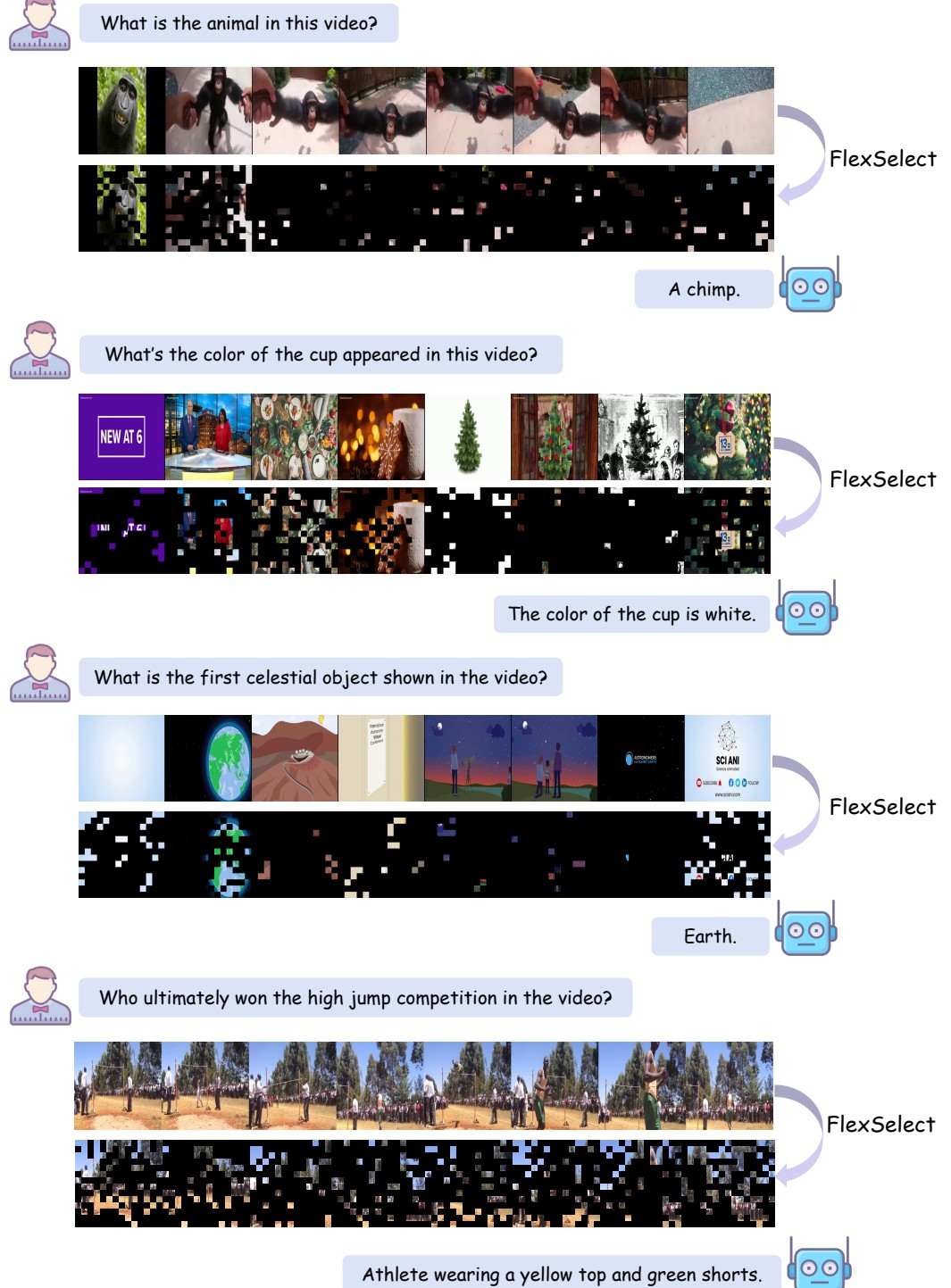

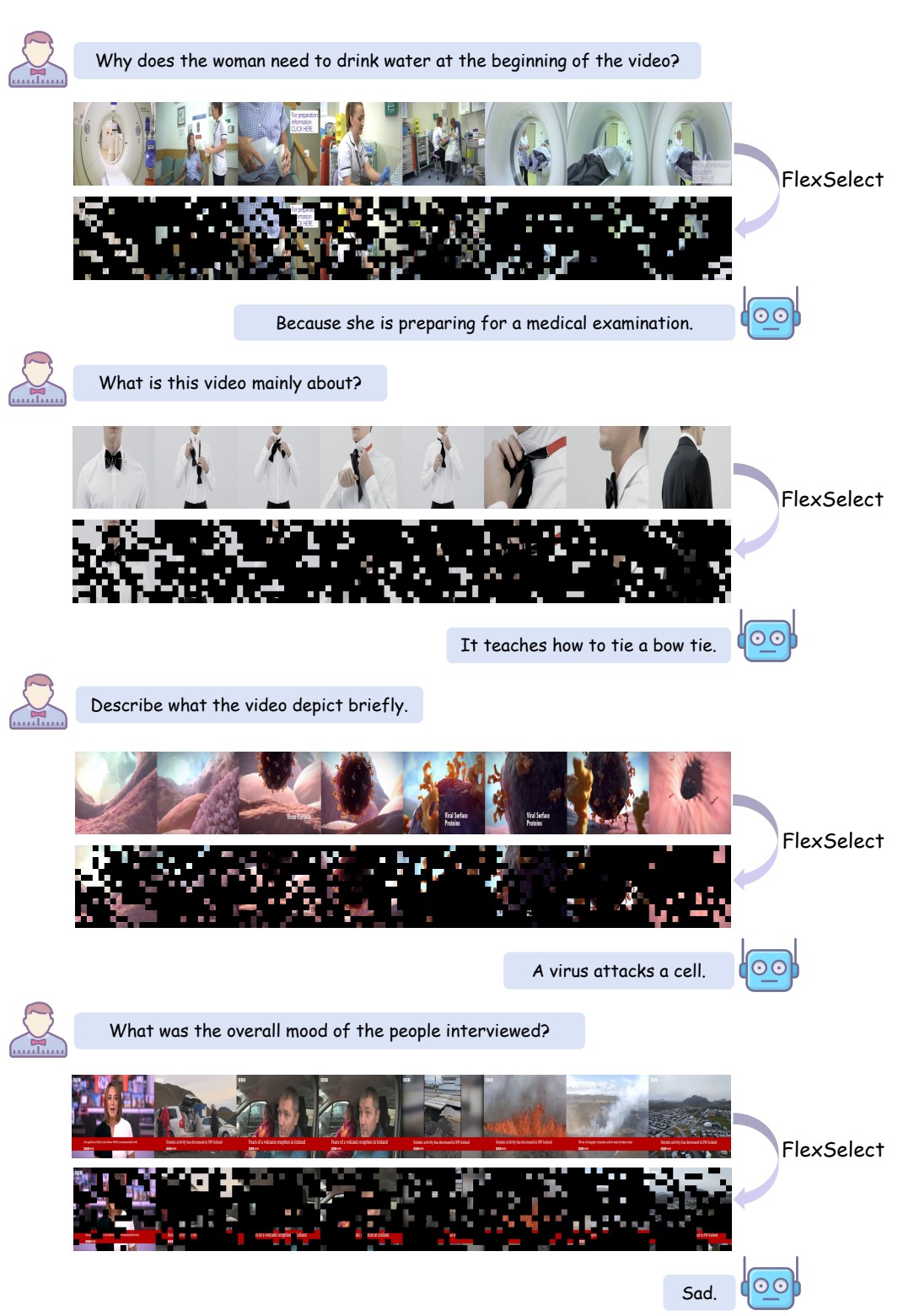

Figure 9: Examples of FlexSelect on Local and Holistic Questions (LLaVA-Video-7B). We choose 8 frames from the sampled frames for visualization.

