# OpenReview forum: "FlexSelect: Flexible Token Selection for Efficient  Long Video Understanding"
_NeurIPS.cc/2025/Conference — NeurIPS 2025 poster_

### Official Review · Reviewer_qmfj · 2025-06-23

**Clarity:** 3
**Significance:** 3
**Originality:** 3
**Rating:** 4
**Confidence:** 4

**Summary:**

This paper introduces FlexSelect, a token selection framework designed to enhance long-form video understanding for Video Large Language Models (VideoLLMs). The method leverages cross-modal attention scores from a reference layer within a pretrained VideoLLM to rank visual tokens by semantic relevance to a given query. It includes two strategies: (1) a training-free selection pipeline based on cross-modal attention scores, and (2) a rank-supervised lightweight token selector trained to mimic the ranking behavior of the full model. Experiments across several VideoLLMs and benchmarks show that FlexSelect achieves consistent accuracy improvements.

**Questions:**

I list my questions and limitations in the weaknesses. My questions are mainly about the conceptual inconsistency in the introduction and the layer selection strategy.

**Ethical Concerns:**

["NO or VERY MINOR ethics concerns only"]

**Final Justification:**

The authors have provided satisfactory empirical evidence and clarifications for most of my concerns in their response. I thus keep the acceptance recommendation.

As a further suggestion, I recommend that the authors provide visualizations of the selected tokens from different layers. This would help qualitatively explore why multi-layer aggregation does not lead to performance improvements, by showing whether the selected tokens across high-performing layers are indeed redundant or highly similar.

**Limitations:**

I list my questions and limitations in the weaknesses. The limitations of this paper lie mainly in the lack of comparison with the referenced attention-based token importance methods [12] and [29], and the potential overfitting to a single layer.

**Paper Formatting Concerns:**

There is no formatting issue.

**Quality:**

3

**Strengths And Weaknesses:**

Strengths:

1. The key insight that semantic alignment peaks at an intermediate layer (identified empirically via Recall@K) is well-validated and intuitive. This forms the foundation for an interpretable and data-efficient ranking mechanism.

2. The approach is architecture-agnostic and plug-and-play. It can be used for various VideoLLMs. The approach design is clear and the paper is well written.

3. Results are presented across multiple models (7B to 72B), datasets, and evaluation protocols, with comprehensive ablations including token limits, layer selection, and training data scale.

Weaknesses:


1. There appears to be a conceptual inconsistency in the introduction. In the second paragraph, the authors state that cross-modal attention scores from pre-trained VideoLLMs often suffer from performance degradation due to inconsistent relevance patterns across transformer layers, making them unreliable indicators of semantic importance. However, in the fourth paragraph, they claim that cross-attention weights between textual queries and visual tokens progressively reflect semantic alignment and typically peak at an intermediate depth—which serves as a key motivation for their method. While it is understandable that the goal is to identify the most semantically informative layer, the narrative should more clearly distinguish between the general limitations of using raw attention and the motivation to selectively leverage it. Moreover, the paper does not include a direct comparison with prior attention-based token importance methods, such as [12] and [29], even though they also experimented on the VideoMME dataset.


2. In Figure 2, several intermediate layers yield similarly high Recall@K scores, suggesting that semantic alignment is not restricted to a single layer. However, the authors use only the layer with the highest score to generate token-level ground truth for supervised training. A more robust strategy could be to aggregate attention scores across multiple top-performing layers to better capture token importance. This could reduce potential overfitting to a single layer and improve the generalizability of the learned token selector

---

> ### Author Rebuttal · Authors · 2025-07-29
>
> We appreciate your valuable comments. We address your concerns below:
>
> **Q1: Narrative should be more clear**
>
> We acknowledge the narrative could be clearer. Our analysis reveals that specific intermediate layers provide faithful semantic alignment, while early and late layers show poor alignment. However, existing methods fail to identify these optimal layers: DyCoke [29] prunes tokens at layer 3, while FrameFusion [12] operates at the initial layers (0, 1, 2). Based on our analysis, attention scores in these early layers are highly unreliable for semantic relevance estimation. As discussed in the fourth paragraph, we observe that semantic alignment between textual queries and visual tokens gradually strengthens with deeper layers, peaking at an intermediate layer (e.g., layer 19 for LLaVA-Video-7B). This finding serves as the key motivation for FlexSelect and directly explains why prior attention-based methods suffer from performance degradation.
>
> **Q2:  Lack of comparison with the referenced attention-based token importance methods  Framefusion [12] and Dycoke [29]**
>
> We appreciate this suggestion and will include direct comparisons in the revision. We compared FlexSelect with FrameFusion [12] on LLaVA-Video-7B and DyCoke [29] on LLaVA-OV-7B, using the same number of sampled frames and pruning ratio. Here are our results.
>
> | Method                      | Sample Frames | Retain Ratio (%)| VideoMME |
> |-----------------------------|--------------|--------------|----------|
> | LLaVA-Video-7B              | 64           | 100.00        | 64.4     |
> | LLaVA-Video-7B+FrameFusion  | 64           | 30.00          | 61.3     |
> | LLaVA-Video-7B+FlexSelect   | 64           | 30.00          | 64.9     |
> | LLaVA-Video-7B+FrameFusion  | 512          | OOM          | OOM      |
> | LLaVA-Video-7B+FlexSelect   | 512          | 6.25        | 68.9     |
>
> | Method               | Sample Frames | Retain Ratio (%)| VideoMME |
> |----------------------|--------------|--------------|----------|
> | LLaVA-OV-7B          | 32           | 100.00        | 58.5     |
> | LLaVA-OV-7B+DyCoke   | 32           | 14.25       | 58.3     |
> | LLaVA-OV-7B+FlexSelect| 32          | 14.25       | 60.4     |
> | LLaVA-OV-7B+DyCoke   | 512          | OOM          | OOM      |
> | LLaVA-OV-7B+FlexSelect| 512         | 6.25        | 63.7     |
>
> The results show that FlexSelect achieves better performance than DyCoKe and FrameFusion under the same number of input frames and retain ratio. Additionally, FlexSelect supports 512 input frames due to its Frame Set Partition mechanism, obtaining more performance improvements.
>
> **Q3: Potential overfitting to a single layer.**
>
> Thanks for the insightful question. We followed your suggestion and conducted additional experiments to investigate potential overfitting to a single layer. We identify layers with high Recall@K scores from Fig. 2 (layers 14, 16, 19, and 21 for LLaVA-Video-7B). We conduct training-free token selection using averaged attention scores and evaluate on the VideoMME benchmark. The results are below:
>
> | Layers       | VideoMME |
> |--------------|----------|
> | 19           | 64.5     |
> | 14+19        | 64.5     |
> | 16+19        | 64.6     |
> | 19+21        | 64.3     |
> | 14+16+19+21  | 64.5     |
>
> The results show that multi-layer aggregation yields comparable performance to single-layer selection, with only marginal differences.
> This suggests two key insights: (1) layers with high Recall@K tend to focus on similar visual tokens with comparable rankings, making aggregation effects minimal, and (2) our single-layer approach is not overfitting but rather efficiently capturing the semantic alignment already present across multiple high-performing layers.
>
> We will add this multi-layer analysis to demonstrate the robustness of our approach.

---

> > ### Comment · Reviewer_qmfj · 2025-08-05
> >
> > Thank you for the response and the additional experiments.
> >
> > Regarding Q1: I appreciate the authors' clarification on the semantic alignment trends across layers and the distinction between unreliable early/late layers and informative intermediate layers.
> >
> > Regarding Q2: I appreciate the authors’ willingness to add direct comparisons in the revised manuscript. The reported results suggesting superior performance of FlexSelect under equivalent settings are convincing.
> >
> > Regarding Q3: The additional experiments on multi-layer aggregation provide useful empirical evidence. However, I would also encourage the authors to provide visualizations of the selected tokens from different layers. This would offer a clearer qualitative understanding of how similar (or different) the token selections are across layers and further support the claim that aggregation yields minimal gains due to redundancy (may be or other).
> >
> >  Overall, the authors have provided satisfactory empirical evidence and clarifications for most of my concerns. I thus keep the acceptance rating.

---

> > > ### Author Response · Authors · 2025-08-05
> > >
> > > Thank you for your thorough review and constructive feedback. We are pleased to hear that our efforts address your concerns. In our revision, we will provide additional visualizations of selected tokens from different layers to more clearly show the differences in token selection across various layers. We sincerely appreciate your consideration and insights. Thank you once again!

---

### Official Review · Reviewer_7tpr · 2025-06-25

**Clarity:** 3
**Significance:** 3
**Originality:** 2
**Rating:** 4
**Confidence:** 5

**Summary:**

The authors propose FlexSelect, a token selection strategy designed to enhance the efficiency and scalability of VideoLLMs in processing long-form videos. The approach leverages cross-modal attention patterns from a reference transformer layer to identify and retain semantically relevant content, while filtering out redundant information. This is achieved through two main components: (1) a training-free token ranking pipeline that uses cross-modal attention weights to estimate token importance, and (2) a lightweight rank-supervised selector trained to replicate these rankings. The proposed method is designed to be modular and can be readily integrated into existing VideoLLM architectures, serving as a plug-and-play solution to extend temporal context handling capabilities. Empirical evaluations demonstrate consistent performance improvements across multiple long-video benchmarks, including VideoMME, MLVU, LongVB, and LVBench. Notably, the approach also yields significant computational speed-ups, highlighting its potential for practical deployment in resource-constrained setting.

**Questions:**

See weakness，my main concerns are as follows:

1. Is there a necessity to adopt an LLM refer layer instead of traditional video-language models for token selection? Currently, there is a lack of convincing experimental evidence to support this approach.

2. In what aspects do the key innovations and insights provided in this paper differ from previous studies?

**Ethical Concerns:**

["NO or VERY MINOR ethics concerns only"]

**Final Justification:**

After careful consideration, I believe the paper deserves a rating above 4, though it does not yet meet the threshold for a 5. If possible, I would assign a 4.5. Overall, I maintain a positive opinion of this work.

**Limitations:**

yes

**Paper Formatting Concerns:**

No paper formatting concerns were identified in this manuscript.

**Quality:**

3

**Strengths And Weaknesses:**

### Strength

1. The approach of partitioning long videos and adopting token selection proposed in this paper has achieved remarkable effectiveness. It significantly reduces the computational complexity while improving the performance of short-video models in processing long videos.
2. The motivation and analysis of this paper are both reasonable and sufficient. The proposed solution also effectively addresses the corresponding issues, featuring a straightforward and easy-to-follow implementation.

### Weakness

1. Whether using an LLM layer as the refer layer outperforms traditional video language models (e.g., Siglip[1], InternVideo2[2]) remains unaddressed, especially when training a weaker student model for ranking—many studies [3, 4] have shown that traditional video embedding models can achieve similar results. The paper lacks relevant performance comparisons.
2. The motivations for segmenting long videos for token reduction and using attention scores from deep LLM layers have been previously proposed in VideoChat-Flash[5[, but this paper omits comparisons and discussions with that work.
3. The core approach of visual token reduction prior to LLM processing has been explored in prior studies (e.g., VisionZip[6]). The paper requires more detailed comparative analyses or experimental result comparisons with these works.
4. The proposed compression method only enables local compression, failing to consider global compression. For instance, if only the final segment of a long video is relevant, the method still retains preceding content unnecessarily.



[1] Sigmoid Loss for Language Image Pre-Training

[2] InternVideo2: Scaling Foundation Models for Multimodal Video Understanding

[3] LVAgent: Long Video Understanding by Multi-Round Dynamical Collaboration of MLLM Agents

[4] LongVU: Spatiotemporal Adaptive Compression for Long Video-Language Understanding

[5] VideoChat-Flash: Hierarchical Compression for Long-Context Video Modeling

[6] VisionZip: Longer is Better but Not Necessary in Vision Language Models

---

> ### Author Rebuttal · Authors · 2025-07-29
>
> We are truly appreciated for your valuable comments. In the following, we provide responses to the concerns.
>
> **Q1: The proposed compression method only enables local compression, failing to consider global compression**
>
> Regarding the concern about local vs. global compression, FlexSelect performs global token selection through our uniform frame set partition strategy (Sec 3.2). Rather than simple chunk-by-chunk segmentation, we partition frames uniformly across the entire video:
>
> For example, if we have 512 frames and want to split them into 8 segments, segment 1 contains frames [0, 8, 16,..., 504]; segment 2 contains frames [1,9, 17,..., 505]; segment 8 contains frames [7, 15, 31,..., 511].
>
> This ensures each segment has a global view of the entire video, allowing relevant tokens from any temporal position (including only the end of the video) to be selected.
>
> We compare our uniform sampling method with the chunk-by-chunk sampling using LLaVA-Video-7B. The results are below:
>
> |Method|Frame Num|Segments Num|VideoMME|
> |-|-|-|-|
> |Chunk-by-chunk|512|8|67.4|
> |Uniform (Ours)|512|8|68.9|
>
> This demonstrates our method's effectiveness in global compression while maintaining computational efficiency.
>
> **Q2: The motivations for segmenting long videos for token reduction and using attention scores from deep LLM layers have been previously proposed in VideoChat-Flash**
>
> We acknowledge VideoChat-Flash as excellent related work and will add proper citations. Here we clarify our key differences:
>
> 1. The segmentation strategy is different. VideoChat-Flash segments videos into **4-frame clips chunk by chunk** and extracts visual features using a video encoder, then employs an MLP to further compress each clip's features into a more condensed representation. This reduces each frame to just 16 tokens, dramatically decreasing the total token count for long videos and improving efficiency. The motivation behind the segmentation is that **the substantial redundant and repetitive information, such as backgrounds and objects present between adjacent frames in natural videos**.
>
>     FlexSelect, however, segments videos to ensure **the context length of each segment within the model’s maximum context limit**, enabling existing VideoLLMs to select relevant visual tokens from hundreds of sampled frames. Meanwhile, as clarified in Q1, FlexSelect employs a **uniform segmentation** to preserve a global view of the entire video within each video clip, which is different from the chunk-by-chunk segmentation in VideoChat-Flash.
>
> 2. The token pruning strategy is different. VideoChat-Flash employs a progressive visual token dropout strategy based on the observation that when an LLM processes long video contexts, it **attends to the entire video in shallow layers while focusing on local details in deeper layers**. It uniformly drops video tokens in shallow layers and selectively drops them based on attention scores in deeper layers, further improving efficiency and slightly enhancing performance.
>
>     However, FlexSelect analyzes the semantic alignment between text and visual tokens across different layers, identifying **an intermediate layer that aligns these two modalities the best**. FlexSelect simply averages attention scores from this layer and drops low-scoring tokens, achieving significant performance improvement.
>
> **Q3: Traditional video model OR LLM layers for token selection? And comparison with VisionZip**
>
> Thank you for the insightful discussion. Both LVAgent and LongVU employ traditional visual encoders to extract frame features and select relevant video regions. LVAgent introduces a novel multi-round framework that enables collaboration between multiple VideoLLM agents, achieving excellent performance in long video understanding tasks. In its perception stage, LVAgent finetunes a CLIP-ASP visual encoder to extract frame features and calculate clip scores to selecte important video segments. Similarly, LongVU uses DINOv2 to extract visual features from sampled frames and reduces temporal redundancy by computing cosine similarity between features to select relevant frames.
>
> LongVU’s core contribution lies in its efficient training on video data—its five-stage training strategy balances efficiency and robust video understanding. LVAgent’s strength comes from its Selection-Perception-Action-Feedback mechanism, where multi-round reasoning and multi-agent collaboration enable effective long video processing. In contrast, FlexSelect focuses on training-freely selecting query-related tokens. We will add these discussions in the revision.
>
> To compare whether LLM layers or traditional visual encoders are more suitable for token selection, we evaluate VisionZip, which also training-freely selects top-k tokens based on attention scores from the visual encoder as dominant tokens and merges the remaining tokens as contextual tokens. While VisionZip did not report results on any video benchmarks, its official GitHub repository provides an implementation for Qwen2.5VL. We compare VisionZip and FlexSelect using Qwen2.5VL-7B under 64 sampled frames.
>
> |Method|Retain Ratio|VideoMME|LongVideoBench|MLVU|
> |-|-|-|-|-|
> | Qwen2.5VL-7B| 100%| 63.6| 59.3| 61.8 |
> | VisionZip| 50% (45% dominant tokens and 5% contextual tokens) | 62.4| 57.9 | 61.5 |
> | FlexSelect| 50%| 62.6| 58.8| 62.4 |
>
> With retaining the same number of visual tokens, FlexSelect continues achieve better performance than VisionZip, indicating that for token-level visual token selection, attention scores from LLM layers is more effective than traditional video models.
>
> **Q4: In what aspects do the key innovations and insights provided in this paper differ from previous studies?**
>
> Thanks for the question. We have discussed the difference between FlexSelect and VideoChat-Flash, LongVU and LVAgent in Q2 and Q3.  Here we clarify our innovations and insights compared with the FastV and VisionZip.
>
> FastV identifies and analyzes the inefficient visual attention phenomena in MLLMs from the perspective of visual tokens' attention proportion relative to all tokens. They find that after the second layer, image tokens garner an average attention score that amounts to only 0.21% of the score attributed to system prompts, while in the initial two layers, this figure reaches 50%. Thus, they drop visual tokens based on the second layer's attention scores, despite efficiency gains, leading to performance degradation. In contrast, FlexSelect analyzes attention scores from the perspective of semantic alignment between visual tokens and user query tokens through our proposed Recall@K experiment, locating the optimal intermediate layer in the LLM. Using the attention scores from this layer to select query-related visual regions achieves better performance.
>
> VisionZip finds that in visual encoder(CLIP or SigCLIP) of MLLMs, only few visual tokens hold higher attention weights while most visual tokens receive very low attention and add significant redundancy. So they locate the dominate visual tokens with high attention scores in visual encoder and merge the other visual tokens. Different from this, FlexSelect analysis the attention scores from LLM layers.
>
> In summary, FlexSelect investigates the cross-modal interaction patterns between text and visual tokens across different layers in Video-LLMs. The results demonstrate that semantic information in visual tokens is not fully resolved in shallow layers but gradually emerges as depth increases, with this phenomenon being most prominent at an intermediate layer. Using the attention scores from this specific layer enables effective selection of query-relevant visual tokens, significantly improving both the efficiency of long-video processing and overall model performance in Video-LLMs.

---

> > ### Comment · Reviewer_7tpr · 2025-08-04
> >
> > Thank you for your detailed and thoughtful response. I appreciate the clarifications and revisions you have provided, which I believe help to further strengthen the overall contribution of the paper. I will maintain my positive rating and will further consider whether an upward adjustment of the score is appropriate.

---

> > > ### Author Response · Authors · 2025-08-05
> > >
> > > Thank you for your thoughtful review and for recognizing our efforts in addressing the clarifications and revisions. We sincerely appreciate your positive feedback and constructive suggestions, which have significantly improved the paper. We are glad to hear that you are considering an upward adjustment of the score—this is very encouraging for our work. We believe the updated version will meet your expectations.

---

### Official Review · Reviewer_xDCz · 2025-06-26

**Clarity:** 3
**Significance:** 3
**Originality:** 2
**Rating:** 4
**Confidence:** 4

**Summary:**

The work introduces a novel way of subsampling visual tokens for speeding up execution of VideoLLM in VQA settings. The authors observe that textual tokens in the query tend to cross-attend only to a small set of visual tokens, while the vast majority receives low or zero attention. Therefore they propose to use this as a ranking signal for identifying the top-K tokens to keep and discard the rest. This technique is particularly effective when applied at a specific layer of the VideoLLM (identified by the authors and architecture dependant), however it still requires to execute part of the VideoLLM at the full sequence length. As an alternative to further speed up computation the authors propose to learn a standalone lightweight LLM to predict the token ranking, this solution brings down computational cost significantly at a small price in terms of quality. The proposal is tested on top of 5 different VideoLLM across 4 video understanding benchmarks and it can consistently improve performance both in terms of accuracy and speed.

**Questions:**

* How is Eq. 1 averaged over all tokens composing the question?

* Have you tried to select only temporally? AKA keyframe rather than sparse tokens?

* How is “No training” implemented in Tab. 3? The tiny LLM should not generate ranking scores without re-training.

* In Tab. 4 did you scale the number of samples seen together with the model parameters? Could the bigger models be undertrained?

**Ethical Concerns:**

["NO or VERY MINOR ethics concerns only"]

**Final Justification:**

The authors have addressed most of my criticisms in the rebuttal and the many new results and comparison help to position this work wrt competitors. For this reason I'm changing from a negative rating to a leaning positive one.

**Limitations:**

yes

**Paper Formatting Concerns:**

* I don’t understand what the left side of Fig. 1 is supposed to show, the four rows look identical to me

**Quality:**

2

**Strengths And Weaknesses:**

## Strengths

+ FlexSelect (the non lite version) is a training free method that promises to be applicable to any off the shelves VideoLLM. The only requirement is identifying which layer is the best to perform the selection which is a one off operation to do. As such the proposal is very flexible and could readily be deployed on any up and coming model in the future.

+ I appreciated the proposal of FlexSelect-Lite as a way of providing additional speedups for a small cost in terms of accuracy. The proposed loss for training is well explained and nicely grounded on differentiable sorting and allows to dynamically select at inference time the computational budget to use since tokens are completely sorted in order of importance. Other methods in the past used binary classification to decide which tokens to keep or drop which would not allow the same flexibility .

## Weaknesses

a. **Lack of comparison to other training free token dropping methods**: the method is not compared to state of the art solutions in the field, so it is hard to quantify the effectiveness of the proposal compared to SOTA. In particular works it might be worth comparing against are [1] for frame selection and [2,3] for token selection. If a comparison is not pertinent it should be explained in the paper why. Currently this is not addressed at all. In particular this work seems in my opinion conceptually very similar to [3] with the addition being performing the pruning at a later layer and the “lite” version of the method.

b. **Cross-attention based pruning**: the author propose to prune tokens based on the aggregated cross-attention scores at a certain layer between query and visual tokens, intuitively this makes sense only for queries that require a “needle-in-haystack” like scenario, it does not for queries that would require to observe the full video, e.g., “list down the actions in this video”. Moreover, following the experimental analysis of [2], the cross-attention pattern that matters more is the one between the output tokens that the model will generate and the visual tokens; these might also change at every decoding iteration. The results from this work are somehow contradicting the results from [2] and, in lack of a proper comparison, as a reviewer I tend to agree more with the analysis provided in [2].

c. **Identification of the layer to use for cross-attention computation**: following on weakness [b] the authors base their motivation and their layer selection on a “needle-in-haystack” problem. I’m wondering how much the findings would hold for a different category of problems, like how the attention pattern looks for a captioning task? If the outcome of the analysis would be very different and the attention would be quite distributed across several more tokens then the findings of the work will hold only for a subcategory of problems and this should be clearly explained in the work, which is instead currently presented as a general solution for any kind of video understanding problem.

d. [minor] **Need to train FlexSelect-Lite**: The lite version of the method needs to be trained on certain data and might underperform when tested on tasks different from the one it has seen while training. Moreover since the Lite model uses a tiny LLM might be more exposed to the risk of overfitting the training distribution and not being able to handle a bigger set of training samples that would improve generalization (see for example Tab. 3 where performance deteriorates moving from 5 to 10% of training data). Having said that, there does not seem to be a negative generalization on the reported results.

e. [minor] **Potentially missing baseline**: Tab. 5 reports increasing computational saving when consuming more and more frames. However I feel there is a crucial difference in the set up between the proposed method and the baseline LLava-Video, namely that for the baseline the frames are processed all at once, while the proposed method processes first independent partitions of the video selecting only a subset of frames and only at a later stage aggregates them. I wonder what the performance difference would be if the same “ensembling like” approach would be applied to LLava-video. So for example instead of processing 512 frames in parallel, the model would be used to process 8 partitions with 64 frames each and answers would be aggregated using some form of majority voting. Time would be 2.37 * 8 = 18.96 second, making the gains from FlexSelect less impressive, and the ensemble might still benefit final performance of the model.

f. [minor] **Fixed token budget**: The method does not incorporate a way of dynamically selecting how many tokens are relevant for the given task. This would be very useful for practical applications.

## References

1. [Liu, Shuming, et al. "BOLT: Boost Large Vision-Language Model Without Training for Long-form Video Understanding." Proceedings of the Computer Vision and Pattern Recognition Conference. 2025.](https://arxiv.org/pdf/2503.21483)

2. [Tao, Keda, et al. "DyCoke: Dynamic Compression of Tokens for Fast Video Large Language Models." Proceedings of the Computer Vision and Pattern Recognition Conference. 2025.](https://openaccess.thecvf.com/content/CVPR2025/papers/Tao_DyCoke_Dynamic_Compression_of_Tokens_for_Fast_Video_Large_Language_CVPR_2025_paper.pdf)

3. [Chen, Liang, et al. "An image is worth 1/2 tokens after layer 2: Plug-and-play inference acceleration for large vision-language models." European Conference on Computer Vision. Cham: Springer Nature Switzerland, 2024.](https://arxiv.org/pdf/2403.06764)

---

> ### Author Rebuttal · Authors · 2025-07-29
>
> We appreciate your valuable comments. Below are our responses:
>
> **Q1: Lack of comparison to other token dropping methods**
>
> We appreciate this suggestion and will include direct comparisons in the revision. We compare the VideoMME result with BOLT, DyCoke and FastV on LLaVA-OV-7B.
>
> For comparison with BOLT, we sample at 1 fps (max to 512 frames) and select 32 * 196 tokens per video, which is the same as BOLT. The result shows that FlexSelect with fine-grained token-level selection significantly surpass BOLT that with frame-level selection.
>
> | | VideoMME|
> |-|-|
> |LLaVA-OV-7B| 58.5|
> |BOLT|59.9|
> |FlexSelect|63.7|
>
> For comparison with DyCoke and FastV, we maintained their experimental settings, sampling 32 frames per video. FlexSelect consistently outperforms both DyCoke and FastV across different retain ratios.
>
> || Retain Ratio (%)| VideoMME wo Sub | VideoMME w Sub |
> |-|-|-|-|
> |LLaVA-OV-7B|100|58.5|61.3|
> |FastV|35|57.3|60.5|
> |DyCoke|14.25| 58.3|60.7|
> |FlexSelect|14.25| 60.4|62.2|
> |DyCoke|18.75| 59.5|61.4|
> |FlexSelect|18.75|61.0|62.4|
> |DyCoke|23.25|58.8|61.0|
> |FlexSelect|23.25|60.4|63.0|
>
> The key difference from FastV is our systematic identification of the optimal reference layer through semantic alignment analysis. FastV considers the proportion of visual tokens' attention scores in the whole attention map, suggesting that layers with higher proportions are more suitable for token pruning. It points out that in MLLM, visual tokens garner an average attention score that amounts to only 0.21% of the score attributed to system prompts, but this figure reaches 50% in the initial two layers. Therefore, they choose to prune tokens at layer 2. FlexSelect considers the semantic alignment between the vision and language modalities, suggesting that layers with attention scores that better locate query-related visual tokens are more suitable for token pruning. Thus, we prune tokens at the reference layer. The experiment results show that our method significantly surpasses FastV.
>
> **Q2: Beyond the NIAH-Type queries**
>
> Thanks for your question.
>
> Our method generalizes beyond NIAH-Type scenarios. It performs well for both NIAH-type and holistic queries. We evaluate FlexSelect on four well-designed long video benchmarks that include diverse types of questions, such as complex reasoning, spatial temporal perception. Our results show that FlexSelect consistently enhances the performance across these tasks, no matter the query is NIAH-type or holistic-type. For example, MLVU benchmark contains Holistic Task that consists of Topic Reasoning (TP) and Anomaly Recognition (AR). These questions require the models to watch the entire video to answer correctly.
>
> ||TP|AR|
> |-|-|-|
> |LLaVA-Video-7B|84.8|66.5|
> |LLaVA-Video-7B+FlexSelect|87.1|70.0|
>
> **Q3: Concerns about contradictions with the results of DyCoke**
>
> Thanks for your question. The experimental analysis in Table 5 of DyCoke [12] shows that cross-attention based pruning (w/o DP) performs worse than the original model, which seems to contradict with our results but actually not. There are crucial differences that explain our divergent findings:
>
> 1. Attention type is different. DyCoke uses output-to-visual attention while we use query-to-visual attention. This is fundamentally different - we analyze attention during the prefilling stage before generation begins, while they analyze attention during the decoding process.
> 2. Layer selection strategy is different. DyCoke performs pruning at layer 3, while we systematically identify and use the optimal reference layer (e.g., layer 19 for LLaVA-Video-7B).
>
> We compared the results of using these two types of attention scores for token pruning at layers 3 and 19, and the results are below. The experiment is conducted under LLaVA-OV-7B with 32 sampled frames. We found that (1) Query-to-visual attention contains more explicit semantic information and consistently outperforms output-to-visual attention. (2) Regardless of the attention type, the performance at layer 3 is worse than the original model, which aligns with FlexSelect's analysis and is consistent with the results in DyCoke.  (3) Our reference layer selection is crucial for achieving performance gains. The dramatic performance difference between layer 3 and layer 19 pruning (56.7→60.4 for query-to-visual) underscores the importance of our systematic approach to identifying the optimal reference layer.
>
> |Prune Layer|Retain Ratio (%)|Cross-Attn Type|VideoMME|
> |-|-|-|-|
> |Layer 19|14.25|output-to-visual|58.1|
> |Layer 19|14.25|query-to-visual|60.4|
> |Layer 3|14.25|output-to-visual|55.8|
> |Layer 3|14.25|query-to-visual|56.7|
> ||||
> |LLaVA-OV-7B|100|-|58.5|
>
> We ensure that our results are reproducible, and will open source all the code related to the results in the future.
>
> **Q4: Caption Task Result**
>
> Thanks for the question, we compare results of VideoDC on LLaVA-OV-7B with 32 sampled frames, and will include it in our revision. Here are the results.
> ||Retain Ratio (%)|VideoDC score|
> |-|-|-|
> |LLaVA-OV-7B|100.00|3.30|
> |LLaVA-OV-7B+FlexSelect|23.25|3.29|
>
> The result shows that FlexSelect achieves comparable performance with the original model. We find that when models are prompted with queries like "Describe the video in detail", the reference layer's attention patterns still exhibit meaningful selectivity, focusing on important elements like main subjects and actions while deemphasizing unnecessary background. The cross-modal attention mechanism inherently adjusts its selection strategy based on query type—focusing narrowly for specific questions while maintaining broader coverage for descriptive tasks. This indicates that FlexSelect's semantic relevance scoring effectively handles various video understanding scenarios.
>
> **Q5: Overfitting and generalization of token selector**
>
> Thanks for the discussion. Our token selector is trained on a randomly sampled subset of LLaVA-Video-178K, which contains diverse task types including QA and captioning.
> The results in Table. 1 show that our token selector (FlexSelect-Lite) can generalize well to different question types.
>
> Regarding model scaling (Table 4), larger token selectors do achieve marginally better performance (67.2→67.4), suggesting improved generalization capacity. However, this comes at significant computational cost (3× response time for 3B vs 0.5B models). Given the minimal performance gains, the 0.5B model offers the optimal efficiency-performance trade-off for practical deployment.
>
> **Q6: Missing baseline: Majority Voting**
>
> We compare majority voting with FlexSelect on LLaVA-Video-7B and will include it in our revision. Here are our results:
>
> ||Sampled Frames In Total|Time Cost|VideoMME|
> |-|-|-|-|
> |LLaVA-Video-7B|64| 2.37s|64.4|
> |Majority Voting|64*8|2.37*8=18.96s|64.9|
> |FlexSelect|512|13.32s|68.9|
>
> For majority voting, we divided each video into 64 temporal bins, randomly sampled one frame per bin, and repeated this process 8 times. The final answer was determined by the most frequent prediction.
>
> FlexSelect significantly outperforms majority voting while being more efficient. It enables the model to reason over semantically relevant tokens from the entire video simultaneously, rather than combining isolated predictions.
>
> **Q7: Dynamic token selection for tasks**
>
> Thanks for your insightful advice. Actually, the number of necessary visual tokens varies by task, with some tasks relying on a few key tokens and others requiring a thorough visual analysis. As shown in table below, for caption tasks (VideoDC), performance improves as more tokens are selected, but this trend does not apply to VideoMME. While adaptively choosing the number of tokens based on user queries is a promising research direction, it poses challenges due to the complexity of the task and the difficulty of the video content. We leave this as the future work.
>
> ||Retain Ratio (%)|VideoDC|VideoMME|
> |-|-|-|-|
> |FlexSelect|14.25|3.11|60.4|
> |FlexSelect|18.75|3.20|61.0|
> |FlexSelect|23.25|3.29|60.4|
> |FlexSelect|50.00|3.30|59.5|
> |LLaVA-OV-7B|100.00|3.30|58.5|
>
> **Q8: How Eq. 1 averaging scores across question tokens**
>
> We average the attention scores between each visual token and all user query tokens across head dimension:
>
>  $r_i = \frac{1}{H \cdot L_t} \sum_{h=1}^{H} \sum_{j=1}^{L_t} attn[h,j,i]$
>
> where H is the number of attention heads; $L_t$ is the number of query tokens; attn is the attention score between the j-th query token and the i-th visual token.
>
> **Q9: Temporal selection only**
>
> Thanks for this discussion. We explored attention based frame selection by averaging the scores (derived from reference layers) across the frame dimension and inference only with the top-k highest-scoring frames. Here are our results on VideoMME using LLaVA-Video-7B.
>
> |Methods|Sample Frames|Selective Frames|Visual Tokens|VideoMME|
> |-|-|-|-|-|
> |Original|128|-|128*210|64.3|
> |Frame Selection|128|64|64*210 (13440)|65.4|
> |Token Selection|128|-|13440|66.1|
>
> The results show that frame selection slightly improves performance but is less effective than token selection when using the same number of visual tokens. We find that this is because most attention scores are nearly uniform and minimal, and some top-scoring visual tokens are irrelevant and act as noise. As a result, the average attention scores at the frame level become smooth, making it hard to identify key frames.
>
> **Q10: "No training" in Table 3**
>
> Thanks for the question. In the "No training" configuration, we directly utilize attention scores from the final layer of the tiny LLM to rank visual tokens without training. Owing to the pre-existing visual knowledge, the tiny LLM could produce a reasonable rank.
>
> **Q11: Were larger models undertrained?**
>
> Thanks for this question. The larger token selector in Table 4 was well-trained. We attempted to train the larger token selector on a larger-scale dataset and found that the loss had already converged, and the performance remained basically unchanged when training with more data.

---

> > ### Comment · Reviewer_xDCz · 2025-08-04
> > **Acknowledgement and follow up**
> >
> > Thanks for the (many) new results! I think they clearly make a better supporting story for the paper and I would suggest to the authors to include them in a revised version of the manuscript. Given the new results I will likelly raise my score as most of the existing doubts and possible weaknesses have been clarified (still need to check in details all other reviews).
> >
> > Regarding Q2 I still think that AR is a somehow NIAH like problem. TR not super sure, but being closed ended might still be a bit biased and solved just by using language priors and a good selection of some relevant frames (no need to check the whole video). This is however not a criticism of this work but more to the MLVU benchmark which is outside the scope of this discussion.
> > The results on captioning benchmarks are instead very promising to show that the method does not work only in a NIAH like scenario.
> >
> > The only remaining concern is that the ratio of selected tokens,, as the authors reported, is a somehow sensible hyperparameter and would need to be tuned per task and model. I don’t see an easy way around it and I don't think this is a concern big enough to suggest rejection.
> >
> > The concern on the generalization of the “lite” version of the model also still intuitively holds somehow, however there is no experimental evidence to support it so the work should not be penalized for it.

---

> > > ### Author Response · Authors · 2025-08-05
> > >
> > > Thank you for your thorough review and constructive feedback. We greatly appreciate your thoughtful comments and are pleased that our response has addressed most of your concerns. We will include these results in our revised version.
> > >
> > > Regarding the NIAH-like tasks, we agree with your observation about potential biases in close-ended questions.
> > > To further demonstrate FlexSelect's effectiveness beyond NIAH scenarios, we evaluated it on the open-ended problems of the VideoEvalPro benchmark, which includes Holistic Perception (HP) and Holistic Reasoning (HR) tasks. HP requires global understanding of statistical, structural, or spatial information through visual aggregation, while HR demands abstract understanding across events or scenes, involving narrative or intent comprehension. Here are our results:
> > >
> > > |  | HP   | HR   |
> > > |--------------------|------|------|
> > > | LLaVA-Video-7B     | 20.7 | 19.3 |
> > > | FlexSelect         | 26.5 | 20.8 |
> > >
> > >
> > > These results demonstrate that FlexSelect consistently improves performance on open-ended holistic queries that require comprehensive video understanding, further validating its generalization beyond NIAH-type scenarios.
> > >
> > > Regarding the token ratio sensitivity, we acknowledge that this remains a key challenge. As we discussed, different tasks require varying amounts of tokens, and determining the appropriate token retention ratio based on both video content complexity and query complexity is non-trivial. While our current approach uses a fixed ratio that works reasonably well across tasks (around 6.25% for 512 frames), we acknowledge that adaptive token selection remains an important direction for future research. Developing methods to dynamically adjust retention ratios based on sample characteristics would significantly enhance the practical applicability of our approach.
> > >
> > >
> > > Regarding FlexSelect-Lite generalization, we have found that the 0.5B token selector performs well in terms of both speed and performance across different tasks, making it the most viable configuration for practical deployment. However, as noted in our Limitations, the rank-supervised training currently cannot fully match the performance of training-free FlexSelect. Exploring ways to enhance the token selector's performance through further effective training or by scaling the token selector's parameter size to potentially surpass training-free methods remains worthwhile.
> > >
> > > Thank you once again for your valuable insights throughout the review process. Your feedback has been instrumental in strengthening our work and clarifying our contributions.

---

> > > > ### Comment · Reviewer_xDCz · 2025-08-05
> > > > **Follow up**
> > > >
> > > > What was the token ratio used for HP and HR in VideoEvalPro?
> > > > And what was the distribution of selected tokens? Do they tend to spread more accross the whole video compared to other tasks?

---

> > > > > ### Author Response · Authors · 2025-08-06
> > > > >
> > > > > Thank you for this insightful follow-up question.
> > > > >
> > > > > For the VideoEvalPro evaluation with LLaVA-Video-7B, we used the same configuration as our main experiments: 512 sampled frames with 6.25% retention ratio.
> > > > >
> > > > > We observed that HP and HR tasks show broader token distribution across the entire video compared to Local Perception (LP) and Local Reasoning (LR) tasks, which focus on specific video segments. This makes sense as holistic tasks require understanding patterns or narratives spanning the full video, while local tasks concentrate on specific moments. For example, given the HR query "What is the audience's attitude towards Yvette and Mario?" for the video vHlSoxg8WHo.mp4, we observe that the selected tokens are distributed relatively evenly across different sampled frames, with a slight concentration on moments when the audience, Yvette, and Mario appear. In contrast, given the LP query "What color is the first plane?" for the video q01CUy_gwdU.mp4, we observe that the selected tokens are clearly concentrated on the frames where the plane appears.
> > > > >
> > > > >
> > > > > We will include visualizations of token distribution patterns across different task types (including captioning) in our revised version to clearly illustrate this.

---

### Official Review · Reviewer_TShu · 2025-07-01

**Clarity:** 3
**Significance:** 2
**Originality:** 2
**Rating:** 4
**Confidence:** 5

**Summary:**

This paper focuses on token selection in long video understanding with LLM. The authors analyze the attention distribution in existing VideoLLMs and find the optimal layer that indicates semantic relevance to guide token selection. The experiments show the efficiency of the proposed strategy.

**Questions:**

Did the authors try soft token merging or adaptive region resize instead of hard token selection through the learned token selector?

**Ethical Concerns:**

["NO or VERY MINOR ethics concerns only"]

**Final Justification:**

The paper provides a view for token selection in video understanding and presents competitive experimental analysis. And more OOD cases would make it more supportive.

**Limitations:**

See weakness

**Quality:**

2

**Strengths And Weaknesses:**

Strength:
1. The semantic token selection significantly reduces the token number and retrains necessary information.
2. The experiments show performance improvements with significantly reduced response latency.

Weakness:
1. The quantitative layer-wise semantic relevance analysis through a needle-in-haystack design is not ideal. The data distribution of the synthesized video is quite different from the natural videos.
2. The semantic salient tokens alone are not sufficient for all questions. Besides, I am curious how will the model select tokens when given a general instruction like captioning.
3. The detailed comparison on the number of selected frames, input tokens, performance and inference latency is desired. And the performance comparison on models with large context window is preferred, e.g., both sampling 512 frames w/ and w/o FlexSelect without exceeding the context window.
4. More visualizations on the selected tokens are required.

---

> ### Author Rebuttal · Authors · 2025-07-29
>
> We appreciate your valuable comments. In the following, we provide responses to the concerns.
>
> **Q1: The Needle-In-Haystack design is not ideal. The data distribution is different from the natural videos.**
>
> Thanks for the question. The Needle-In-Haystack task is a well-established benchmark in video understanding. This task assesses whether a Video-LLM can identify visual cues from the extensive visual tokens based on the query, and has been used in recent works such as the NQA (Needle QA) test in MLVU, the V-NIAH experiment in LongVA, and Multi-Hop V-NIAH in VideoChat-Flash. We choose this task because it allows us to more clearly and easily observe the alignment between text and visual tokens across different layers.
>
> To address your concern, we conducted additional experiments on natural videos using the temporal grounding task. We randomly sampled 128 videos from Charades-STA, where each video is paired with a ground-truth temporal range relevant to one question. We treated tokens within the ground-truth range as semantically relevant and others as irrelevant, then computed the layer-wise Recall@K for LLaVA-Video-7B. Here are our results:
>
> |Layer|Recall@K|Layer|Recall@K|Layer|Recall@K|Layer|Recall@K|
> |-|-|-|-|-|-|-|-|
> |0|0.2086|7|0.3092|14|0.3679|21|0.3638|
> |1|0.2587|8|0.3336|15|0.3155|22|0.3686|
> |2|0.2734|9|0.3423|16|0.3711|23|0.2641|
> |3|0.2790|10|0.3280|17|0.3344|24|0.2391|
> |4|0.2626|11|0.3448|18|0.2867|25|0.2541|
> |5|0.2996|12|0.3191|19|**0.3688**|26|0.2632|
> |6|0.3035|13|0.3021|20|0.3228|27|0.2385|
>
>
>
> The results on natural videos show a consistent pattern with our synthetic experiments: Recall@K increases through early layers, peaks at intermediate layers (layers 16, 19, 22), and declines in deeper layers. This validates that our reference layer selection methodology generalizes to real-world video data.
>
>
> **Q2: The semantic salient tokens alone are not sufficient for all questions. What about the result of caption tasks.**
>
> Thanks for your question. We acknowledge that different task types may require varying levels of visual information. Our experiments demonstrate that FlexSelect adapts effectively across diverse tasks, including captioning.
>
> In Table. 1, we evaluated FlexSelect on four long video benchmarks (VideoMME, LongVideoBench, MLVU, and LVBench) that include diverse types of questions, such as complex reasoning, spatial temporal perception. Our results demonstrate that FlexSelect consistently enhances the performance of VideoLLM across these tasks.
>
> To specifically address captioning, we evaluated FlexSelect on the VideoDetailCaption (VDC) benchmark using LLaVA-OV-7B with 32 sampled frames. Retain ratio refers to the keep ratio of visual tokens.
>
> |Method|Retain Ratio (%)|VDC score |
> |-|-|-|
> | LLaVA-OV-7B (Baseline)|100.00|3.30|
> | FlexSelect|14.25|3.11|
> |FlexSelect|18.75|3.20|
> |FlexSelect|23.25|3.29|
> |FlexSelect|30.00|3.30|
>
> The results reveal that captioning tasks benefit from higher token retention ratios. With 23.25% retention, FlexSelect achieves comparable performance (3.29 vs 3.30) to the baseline while also improves efficiency a lot.
>
> Interestingly, even for open-ended queries like "Describe the video in detail" the reference layer's attention patterns still exhibit meaningful selectivity. The model naturally focuses on salient elements (e.g., main subjects, significant actions) while deemphasizing redundant background information. For example, when describing the video in Fig. 3, the model attends to relevant elements like the commentator and Christmas decorations while filtering repetitive backgrounds. The cross-modal attention mechanism inherently adjusts its selection strategy based on query type—focusing narrowly for specific questions while maintaining broader coverage for descriptive tasks. This demonstrates that FlexSelect's semantic relevance scoring is sufficiently flexible to handle diverse video understanding scenarios.
>
> **Q3: The detailed comparison on the number of selected frames, input tokens, performance and inference latency is desired**
>
> Thank you for your advice. The effects of the input frame number and selected token number on performance and time cost have been analyzed in Table 2 and Figure 5. Here we provide a more detailed comparison. The tables below show the VideoMME results and time cost at different input frames and selected tokens.
>
> | |64 Frames|128 Frames|256 Frames|512 Frames|1024 Frames|
> |-|-|-|-|-|-|
> |1,680 Tokens|64.5|66.2|66.6|67.1|66.6|
> |3,360 Tokens|64.7|66.9|67.0|68.4|68.1|
> |6,720 Tokens|65.2|67.8|68.1|68.9|68.1|
> |13,440 Tokens|64.4|66.1|66.5|68.1|67.0|
>
>
> | |64 Frames|128 Frames|256 Frames|512 Frames|1024 Frames|
> |-|-|-|-|-|-|
> |1,680 Tokens|1.95s|3.63s|6.55s|12.82s|26.84s|
> |3,360 Tokens|2.01s|3.95s|6.86s|13.10s|27.40s|
> |6,720 Tokens|2.24s|4.64s |7.38s|13.32s|27.58s|
> |13,440 Tokens|2.37s|5.54s|8.54s|24.87s|29.32s|
>
> Our experiments show that using 512 input frames with 6,720 selected tokens (6.25% retention) achieves 68.9% accuracy on VideoMME, improving from the original model's 64.4%. This supports our hypothesis that semantic token selection enhances efficiency and accuracy in long-form video understanding.
>
> Additionally, FlexSelect enables processing up to to 1,024 frames—avoiding OOM errors in the baseline—while maintaining a competitive 68.1% accuracy, offering crucial scalability for real-world applications needing extended temporal coverage.
>
> **Q4: The performance comparison on models with large context window is preferred, e.g., both sampling 512 frames w/ and w/o FlexSelect without exceeding the context window.**
>
> Thanks for the question. Qwen2.5VL series models support long window context. Following the official repository, we use YARN with scaling factor = 4.0 to expand the context window length from 32k to 128K. The result of Qwen2.5VL (original) we reported in the main table was tested under 768 sampled frames (about 107k tokens). Here are the results:
>
> |Model|Sample frames |Visual token number | VideoMME |LongVB|MLVU| LVBench |
> |-|-|-|-|-|-|-|
> |Qwen2.5VL|768| About 107k|65.4|59.5|70.2|45.3|
> |Qwen2.5VL (FlexSelect)|1024|7010|68.2|62.4|72.5| 51.2|
>
> The results show that FlexSelect with just 7,010 tokens outperforms the baseline with 107k tokens, achieving higher scores on all four benchmarks. By focusing on semantically relevant tokens, FlexSelect reduces the noise and redundancy inherent in dense video representations and improves performance, independent of the model's context window length.
>
> **Q5: Did the authors try soft token merging or adaptive region resize instead of hard token selection through the learned token selector?**
>
> Thanks for the insight discussion. We explored both soft token merging and adaptive region resizing as alternatives to hard token selection, though neither achieved comparable results to FlexSelect.
>
> For soft token merging, we implemented a LinVT-inspired approach on InternVL2-2B, which trains an adapter to compress visual tokens into fixed number of token sets before they are fed into LLMs. The adapter consists of a Spatio-temporal Visual token Refiner module (SVR) to distill the video information into a concise set of visual tokens, and a Text-conditioned Token Aggregator (TTA) module to refine and integrate the visual tokens with the textual information. In our implementation, we use 4 full transformer layers as TTA and 8 spatial-temporal transformer layers as SVR.
>
> The parameters of the adapter are randomly initialized and we trained it in two stages on video instruction data. The first stage only trains the adapter and the second stage trains both the adapter and the LLM.  We finally found that such training degraded the performance.
> Here are our results:
>
> ||Frame Number| Token Number  |VideoMME |
> |-|-|-|-|
> |Original model| 32| 32*256=8192| 51.5|
> |Soft Token Merging Training| 32| 1024| 50.5|
>
> We hypothesize that learning-based compression introduces information loss that cannot be recovered, whereas FlexSelect preserves the critic information well by selecting semantically relevant visual tokens.
>
> For adaptive region resizing, we explore a similar training free method on InternVL2.5-8B. The visual encoder of InterVL2.5-8B supports both of 224 * 224 and 448 * 448 resolution inputs. We first resize the video frames into the small resolution and locate the important frames based on the query-to-visual relevance scores. And then we resize the important frames to the higher resolution while remains the other frames as low resolution. Here are our results:
>
> || Sample Frame Num |High Res Frame Num| Visual Token Num | VideoMME |
> |-|-|-|-|-|
> | original model| 64 | 64 | 16384| 64.2|
> | adaptive resize| 64| 16| 7168| 62.4|
> | adaptive resize| 64| 32| 10204| 63.7|
> | adaptive resize| 128 | 32| 14336| 64.2|
> | FlexSelect | 64 | - | 2458 | 65.6|
>
> Results showed consistent underperformance: with 64 frames (16 high-res), accuracy dropped from 64.2% to 62.4%; even with 128 frames (32 high-res), we only matched baseline performance while using 14,336 tokens, while FlexSelect's 2458 tokens achieving 65.6%.
>
> **Q6:More visualizations on the selected tokens are required.**
>
> Thank you for this suggestion. In our revised version, we will include more comprehensive visualizations to fully demonstrate FlexSelect's robustness and adaptability across different tasks.
>
> Besides the visualizations in Appendix A.6 (Figure 9), we observe consistent patterns in different question types: (1) Object-specific queries focus token selection around relevant objects; (2) Action queries select tokens across motion sequences with temporal coherence; (3) Counting queries distribute selection to capture all instances. These visualizations show our cross-modal attention mechanism effectively adapts to query semantics.
>
> **References:**
> 1. [LinVT: Empower Your Image-level Large Language Model to Understand Videos](https://arxiv.org/abs/2412.05185)

---

> > ### Comment · Reviewer_TShu · 2025-08-05
> >
> > Thanks for the rebuttal. The new results have addressed most of my concerns. And the paper would benefit from providing more OOD cases in the revised version. I will raise my rating.

---

> > > ### Author Response · Authors · 2025-08-05
> > >
> > > Thank you for your thoughtful review and for recognizing our efforts to address your concerns. We are glad to hear that you will raise the rating-this is very encouraging for our work. We will include additional visualizations of OOD cases such as the caption task in our revision. We sincerely appreciate your consideration and insights. Thank you once again!

---

### Decision · Program_Chairs · 2025-09-17

**Decision:**

Accept (poster)

**Comment:**

This paper received four borderline accepts as final ratings. Initially, reviewers praised the paper for its motivation, strong performance of the method, and plug and play nature of the model. However, reviewers had concerns with comparison to other state of the art methods and how this paper builds upon existing work. The rebuttal gave a great deal of information and answered many of the reviewers' questions leading to all positive final ratings. Because of this, the AC sees no reason to overturn the majority decision of the reviewers and recommends acceptance for this paper.

The AC reminds the authors to make the changes that the reviewers proposed and they promised as part of the rebuttal in the camera ready version of the paper.